# Skeletal muscle stem cells modulate niche function in Duchenne muscular dystrophy mouse through YY1-CCL5 axis

Yang Li [1,2], Chuhan Li[1], Qiang Sun[1,2], Xingyuan Liu[3], Fengyuan Chen[3], Yeelo Cheung[1], Yu Zhao[4], Ting Xie[5], Bénédicte Chazaud[6], Hao Sun [7] ✉ & Huating Wang [1,2] ✉

Adult skeletal muscle stem cells (MuSCs) are indispensable for muscle regeneration and tightly regulated by macrophages (MPs) and fibro-adipogenic progenitors (FAPs) in their niche. Deregulated MuSC/MP/FAP interactions and the ensuing inflammation and fibrosis are hallmarks of dystrophic muscle. Here we demonstrate intrinsic deletion of transcription factor Yin Yang 1 (YY1) in MuSCs exacerbates dystrophic pathologies by altering composition and heterogeneity of MPs and FAPs. Further analysis reveals YY1 loss induces expression of immune genes in MuSCs, including C-C motif chemokine ligand 5 (*Ccl5*). Augmented CCL5 secretion promotes MP recruitment via CCL5/C-C chemokine receptor 5 (CCR5) crosstalk, which subsequently hinders FAP clearance through elevated Transforming growth factor-β1 (TGFβ1). Maraviroc-mediated pharmacological blockade of the CCL5/CCR5 axis effectively mitigates muscle dystrophy and improves muscle performance. Lastly, we demonstrate YY1 represses *Ccl5* transcription by binding to its enhancer thus facilitating promoter-enhancer looping. Altogether, our study demonstrates the critical role of MuSCs in actively shaping their niche and provides novel insight into the therapeutic intervention of muscle dystrophy.

Skeletal muscle has a robust regenerative capacity, with rapid re-establishment of full power occurring even after severe damage that causes widespread myofiber necrosis[1,2]. The cells responsible for muscle regeneration are adult muscle stem cells (MuSCs, also called satellite cells) which are located in a niche beneath the ensheathing basal lamina on the surface of the myofibers in a quiescent stage under normal conditions[3–5]. Upon injury, MuSCs are rapidly activated to become myoblasts, undergo proliferative expansion, and eventually differentiate into myotubes and fuse to form new myofibers[3–5]. A subset of MuSCs undergoes self-renewal and returns to the quiescent state to replenish the adult stem cell pool[6–8]. Each phase of the activities is tightly orchestrated at two levels. First, through intrinsic pre-programmed mechanisms and, second, through extrinsic regulations imposed by the stem cell microenvironment or so-called stem cell

[1]Department of Orthopaedics and Traumatology, Li Ka Shing Institute of Health Sciences, Chinese University of Hong Kong, Hong Kong SAR, China. [2]Center for Neuromusculoskeletal Restorative Medicine Limited, Hong Kong Science Park, Hong Kong SAR, China. [3]Department of Chemical Pathology, Li Ka Shing Institute of Health Sciences, Chinese University of Hong Kong, Hong Kong SAR, China. [4]Molecular Cancer Research Center, School of Medicine, Shenzhen Campus of Sun Yat-sen University, Sun Yat-sen University, Shenzhen, China. [5]Center for Tissue Regeneration and Engineering, Division of Life Science, Hong Kong University of Science and Technology, Hong Kong SAR, China. [6]Unité Physiopathologie et Génétique du Neurone et du Muscle, UMR CNRS 5261, Inserm U1315, Université Claude Bernard Lyon 1, Lyon, France. [7]Warshel Institute for Computational Biology, Faculty of Medicine, Chinese University of Hong Kong (Shenzhen), Guangdong, China. ✉e-mail: haosun@cuhk.edu.hk; huating.wang@cuhk.edu.hk

niche[9,10]. The intrinsic regulatory mechanisms have been relatively well defined. For example, it is widely accepted that gene regulation at transcriptional level by transcription factors (TFs) plays crucial roles in instructing MuSC regenerative responses[9,11]. Among many key TFs, Yin Yang 1 (YY1) is ubiquitously expressed but possesses unique transcriptional regulatory functions in MuSCs based on findings from our group and others[12–17]. For example, we recently elucidated the function of YY1 in acute injury-induced muscle regeneration as a key regulator of MuSC activation/proliferation. We found that intrinsic YY1 deletion in MuSCs impairs muscle regeneration by regulating MuSC activation/proliferation through its dual roles in modulating metabolic pathways[12]. Moreover, we also observed that the deletion in MuSCs in chronically injured muscle of *mdx* (a mouse model for Duchene muscular dystrophy, DMD) aggravates muscle dystrophy but the in-depth dissection is yet performed[12].

MuSC functionality is also tightly controlled by the crosstalk between MuSCs and other cell types within their niche[18]. The MuSC niche is relatively static under homeostatic conditions to maintain MuSC quiescence and undergo dynamic remodeling following injury through a spatiotemporally tightly coordinated flux of different cell types such as inflammatory, vascular, and mesenchymal cells[10,19]. The reciprocal functional interactions among these cell types are crucial in coordinating the repair of injured muscles. In particular, optimal regeneration entails a sequence of events that ensure temporally coordinated interactions among MuSCs, macrophages (MPs), and fibro-adipogenic progenitors (FAPs)[20–22]. An initial recruitment of MPs is typically followed by the sequential activation of FAPs and MuSCs. The key immune and non-immune functions that MPs exert in muscle regeneration are well accepted[23,24]. The pro-inflammatory response following acute muscle injury usually begins with infiltration of neutrophils, followed by pro-inflammatory MPs featured by Ly6C$^{high}$F4/80$^{low}$, which produce mostly inflammatory cytokines, such as TNFα, IL-1β, and IFNγ to promote MuSC activation and proliferation. Afterward, at later stages of regeneration, a distinct population of Ly6C$^{low}$F4/80$^{high}$ pro-regenerative (anti-inflammatory) MPs become more prevalent, which produce different cytokines, such as IL-4, IL-10 and TGFβ1, to promote myoblast differentiation and tissue repair[23,25]. Although much is known about the impact of MPs on MuSCs, the possible reciprocal regulation of MuSCs on MPs remains largely unexplored despite a pioneer work long time ago demonstrating that human myoblasts can indeed secrete an array of chemotactic factors to initiate monocyte chemotaxis in culture[26]. FAPs are a muscle interstitial mesenchymal cell population which can differentiate into fibroblasts, adipocytes, and possibly into osteoblasts and chondrocytes. Emerging evidence solidifies FAPs' critical role in efficient muscle repair[21]. Upon muscle injury, FAPs become activated, proliferate and expand, and provide a transient favorable microenvironment to promote MuSC-mediated regeneration. It is now known that heterogenous FAP subsets have distinct functions in muscle regeneration. For example, Wisp1+ FAPs expansion is critical during regeneration to sustain MuSC proliferation and differentiation in a paracrine manner and maintain the MuSC pool[27] while mTie2$^{High}$/mTie2$^{Low}$ FAPs play a role in promoting angiogenesis[28]. As regeneration proceeds, the timely removal of FAPs from the regenerative niche through apoptosis is necessary to prevent pathological accumulation and muscle dysfunction. Accumulated fibrogenic FAPs lead to fibrosis[29,30] and adipogenic FAPs result in fat deposition[31]. Again, it remains largely unknown if FAPs receive any reciprocal signaling from MuSCs. Nevertheless, it is widely accepted that finely tuned molecular interactions of FAPs and MPs are essential for successful regeneration. Expansion and decline of FAP cell populations following injury are determined by MPs and disrupted MP dynamics can result in aberrant retention of FAPs in muscles following acute or chronic injuries[20,21]. For example[29], in acutely damaged skeletal muscle, infiltrating MPs, through their expression of TNFα, directly

induce apoptosis and the timely clearance of FAPs, thus preventing the occurrence of fibrosis.

Therefore, the finely orchestrated functional interactions among MuSCs, MPs and FAPs are crucial to instruct the proper progress of damage-induced muscle repair. Conditions that compromise the functional integrity of this network can skew muscle repair toward pathological outcomes, for example in the case of DMD[20,22], which is a lethal progressive pediatric muscle disorder caused by the genetic mutations in the dystrophin gene. In the absence of dystrophin protein, DGC (dystrophin-glycoprotein complex) assembly on the muscle member is impaired which weakens the muscle fibers, rendering them highly susceptible to injury. Muscle contraction-induced stress results in constant cycles of degeneration and regeneration, resulting in chronic inflammation and fibro-fatty tissue replacement. Recent advances in single cell RNA-sequencing (scRNA-seq) have enabled us to characterize cell compositions and interactions in both human and mouse DMD muscles[32,33]. Compared to muscles underlying acute injury-regeneration, more intricate cellular dynamics and interactions are observed in dystrophic muscles. An increased prevalence of MPs and FAPs was confirmed and correlated with disease severity[29,34,35]. MPs are highly heterogeneous and partially pathogenic in dystrophic muscle[36]. A plethora of MP-derived factors are critically involved in inflammation and fibrosis of the muscle. It is speculated that the heterogeneous and pathogenic activation of MPs in dystrophic muscle is likely induced by the asynchronous regeneration altered microenvironment but the underlying signaling molecules and cellular sources remain unexplored[37]. Evidence from the scRNA-seq profiling also revealed evident heterogeneity of FAPs and their correlation with disease severity[32]. FAPs undergo uncontrolled expansion and resistance to clearance in dystrophic muscle, which plays a prominent role in intramuscular fat deposition and fibrosis[38]. Increased TGFβ signaling is believed to prevent FAP apoptosis and induce their differentiation into matrix-producing cells to cause fibrosis[29]; the main source of TGFβ1 in injured muscle niche is MPs. While pro-regenerative MPs are predominantly contributor to TGFβ1 secretion in acutely injured muscle niche, most MPs in chronically injured muscle can secret TGFβ1 to foster a TGFβ1 enriched microenvironment[29].

Overall, cellular communications among the main pathogenic cells in dystrophic muscle warrant further investigation; in particular, it is unknown if MuSCs play an active role in initiating crosstalk with MPs and FAPs to manipulate their own niche microenvironment. In fact, it is safe to state that in general our understanding about if and how tissue adult stem cells impact their niches is very limited. Across a variety of well-studied adult stem cells such as mesenchymal stem cells, hematopoietic stem cells, and neural stem cells etc., we have learned plenty about the essential roles of the stem cell niche in regulating stem cell behavior and functionality[39], including how alteration in the stem cell niche causes cellular damage and impairs the regenerative capacity of stem cells. In principle the stem cells are perceived as the passive recipient of niche signals and impact. There is relatively less understanding of whether and how stem cells can actively contribute to the niche integrity in homeostasis and how the intrinsic changes in stem cells are connected to extrinsic niche alterations in pathological conditions. Here in this study, we discovered that intrinsic deletion of YY1 in MuSCs of dystrophic *mdx* mice (dKO) exacerbated fibrosis and inflammation. Analysis of cellular compositions uncovered elevated numbers of MPs and FAPs accompanying a decrease in MuSCs in dKO vs. control mice; moreover, scRNA-seq profiling revealed altered cellular heterogeneity of MPs and FAPs. Furthermore, we found that YY1 deletion in MuSCs induced up-regulation of immune genes thus rendering MuSCs immunogenic. Notably, CCL5 was identified as a critical factor in facilitating the recruitment of MPs via the CCL5/CCR5 axis mediated MuSC-MP interaction; Escalated MP accumulation subsequently prevented FAP apoptosis and clearance via increased TGFβ1 accumulation. Consistently, treatment of the dKO mice with a CCR5

antagonist Maraviroc (MVC) significantly ameliorated the dystrophic pathologies and muscle function in dKO mice. Lastly, we elucidated that *Ccl5* induction in dKO MuSCs resulted from an altered enhancer-promoter loop interaction. Altogether our findings demonstrate an active role of MuSCs in orchestrating cellular interactions with other niche cells and highlight their capacity in modulating niche microenvironment through their immune-secretory function.

## Results

### Inducible deletion of YY1 from MuSCs aggravates muscle dystrophy in *mdx* mouse

Our prior study[12] has hinted the possible YY1 involvement in chronic degeneration/regeneration occurring in dystrophic *mdx* mouse, which promotes us to further investigate the functional and mechanistic role of YY1 in dystrophic muscle in this study. We conducted a comprehensive characterization of the phenotypes of the *Yy1/mdx* double knockout (dKO) mice that were generated by inducible deletion of YY1 in MuSCs of *mdx* mice (Supplementary Fig. 1A). Five days of consecutive intraperitoneal (IP) administration of tamoxifen (TMX) was performed in 1.5-month-old (1.5 M) Ctrl and dKO mice to induce YY1 deletion in MuSCs and the muscles were harvested when mice were 2.5 M, 3.5 M and 8.5 M old for subsequent analyses (Fig. 1A–D). Consistent with our prior observations[12], YY1 deletion caused severe impairment in muscle regeneration in both limb (tibialis anterior, TA) and diaphragm (DP) muscles as evidenced by the abnormal fiber size distribution with increased larger fibers in the dKO muscles as assessed by H&E (Fig. 1E) and eMyHC (Supplementary Fig. 1B, C) staining at 2.5 M. Moreover, the dKO muscles showed a significantly decreased number of fibers with centrally located nuclei (CLN) (Fig. 1F) and a reduced number of fibers per area (Fig. 1G). Additionally, a decreased number of MuSCs (Supplementary Fig. 1D) caused by impaired proliferation (measured by in vivo EdU incorporation assay in Supplementary Fig. 1E) was also confirmed. Moreover, the dKO mice displayed striking fibrotic- and immune-phenotypes. An excessive amount of fibrosis was detected in dKO by both Masson's trichrome staining (Fig. 1H) and increased expression of a panel of fibrotic markers (Supplementary Fig. 1F, G). The level of fibrosis appeared to arise from the increased number of PDGFRα + FAPs as a positive correlation was detected between the main fluorescence intensity (MFI) of PDGFRα and COL1a1 protein staining in both TA (Fig. 1I) and DP (Fig. 1J) of Ctrl and dKO mice. This was accompanied by an elevated level of inflammation as indicated by increased expression of inflammatory markers (Supplementary Fig. 1H, I) and an augmented number of macrophages (Fig. 1K, L, Supplementary Fig. 1J, K). Altogether, the above phenotypic characterization demonstrates that YY1 deletion in MuSCs leads to exacerbation of dystrophic pathologies in *mdx* mice. The phenotypes persisted in dKO mice into elder age by contrasting with the Ctrl *mdx* mice alleviated at around 3.5-month-old (Supplementary Fig. 1L–Q). As a result, the dKO mice were overall very fragile displaying much smaller body size (Fig. 1M, 8.5 M) and significantly lower body weight and muscle weight (Fig. 1N, 3.5 M); the TA muscle mass decreased by 50.97% (Fig. 1O) and the thickness of DP muscle shrunk by 39.51% (Fig. 1P). When the 3.5-month-old mice were subject to treadmill (Fig. 1Q) or voluntary wheel-running (Fig. 1R) exercises, these dKO mice exhibited poor performance compared to the Ctrl mice. As expected, the survival rate of the dKO mice was significantly reduced and half of them died before 6 months (Fig. 1S). Therefore, the YY1 loss in MuSCs worsens the dystrophic manifestations in the *mdx* mouse.

### Intrinsic deletion of YY1 in MuSCs alters their niche in dystrophic muscle

We hypothesized that YY1 deletion in MuSCs exacerbates dystrophic phenotypes possibly by increasing the numbers of MPs and FAPs in the niche microenvironment while shrinking the MuSC pool. To test this

hypothesis, MuSCs, FAPs and MPs were isolated from limb muscles at 5, 21, and 60 days after TMX administration (Fig. 2A) following established FACS sorting protocols[12,40,41] (Supplementary Fig. 2A–C). As expected, we observed a progressive decline of MuSCs in the dKO but not in the Ctrl (Fig. 2B). Meanwhile, FAPs (Fig. 2C) and MPs (Fig. 2D) were increased at later stages (21- and 60-days post TMX injection), particularly at 21 days. The above results suggest that the intrinsic deletion of YY1 in MuSCs leads to skewed cellular composition in *mdx* mouse muscle.

To further illuminate the altered muscle niche we performed single cell (sc) RNA-seq. As shown in Fig. 2E, hind limb muscles were collected from three pairs of Ctrl or dKO mice (1 month after TMX injection); a mixed single-cell suspension from the three mice was subject to droplet-based scRNA-seq on a 10x Chromium platform. After filtering (cells with fewer than 200 genes and 1000 unique molecular identifiers (UMIs) detected or with more than 5% of UMIs mapped to mitochondrial genes, Supplementary Fig. 2D–H) and doublet removal (Supplementary Fig. 2I), 4335 and 3329 cells were obtained from Ctrl and dKO groups (Supplementary Fig. 2J) for subsequent analyses. A comprehensive definition of cellular atlas uncovered a total of 14 different cell populations including monocyte/macrophage (MO/MP), FAP, endothelial cell (EC), tenocyte, T cell (TC), B cell, smooth muscle cell, dendritic cell, MuSC, neutrophil, cycling basal cell (CBC), myocyte, pericyte and Schwann cell (Fig. 2F) based on normalized gene expression levels and canonical cell type-specific markers (Fig. 2G, Supplementary Fig. 1K, and Supplementary Data 2). Notably, FAP, MO/MP were the major cell populations in the niche, making up over 60% of the total cells in both Ctrl and dKO (Fig. 2H). Consistent with the results from Fig. 2B, C, a significantly increased population of FAPs was detected in dKO vs. Ctrl (35.9% vs. 26.4%) accompanied by a reduced population of MuSCs (1.2% vs. 3.3%). Interestingly, no obvious increase in the number of MO/MP was observed (38.0 % vs. 38.2%). The ratio of most cell populations, including EC (7.2% vs. 12.6%), neutrophil (1.1% vs. 2.6%), and pericyte (0.9% vs. 2.1%), displayed a decrease in dKO vs. Ctrl (Fig. 2H, Supplementary Data 2).

To further illuminate the dynamic shifts of FAPs, pseudotime trajectory was utilized to reveal the cellular heterogeneity and fate determination. Unbiased SNN clustering[42] uncovered five subpopulations of FAPs in Ctrl group and designated as 0-Activated, 1-Stressed, 2-Fibrogenic, 3-Adipogenic and 4-Transitional according to differentially expressed genes (DEGs) (Fig. 2I, Supplementary Fig. 2L, Supplementary Data 2). Activated FAPs localized toward the starting point of the trajectory, which were marked by the expression of *Cxcl5*, *Cxcl3*, *Ccl7*, and *Ccl2*; Stressed FAPs did not appear to have any spatial bias, and were enriched with stress-responsive genes such as heat shock genes (Hspb1, Hspd1, Hspe1) and AP-1 family transcription factors (*Fos*, *Atf3*, *Jun*); Fibrogenic (enriched with ECM factors, *Cxcl14*, *Lum*, *Smoc2*, *Podn*) and Adipogenic FAPs (enriched with adipogenic factors, *Pi16*, *Dpp4*, *Igfbp5*) (Supplmentary Fig. 2L) highlighted the main divergent fates of two FAP subpopulations in the trajectory. Transitional FAPs appeared on the path from the Activated to Fibrogenic/Adipogenic FAPs, marked with *Apod*, *Ptx3*, *Myoc* and *Mt1* genes. As a comparison, above-described five subsets of FAPs were also identified in dKO muscles including 0-Activated, 1-Stressed, 2-Fibrogenic, 3-Adipogenic and 4-Inflammatory (Fig. 2J, Supplementary Fig. 2M, and Supplementary Data 2). And a similar trajectory was adopted: starting from the Activated state, dKO FAPs were arranged along a trajectory that diverged into two distinct branches, which coincided with the two subpopulations, Fibrogenic and Adipogenic (Fig. 2J). Notably, the proportion of the Activated subset increased significantly in dKO vs. Ctrl (32.1% vs. 15.1%), while the Stressed subset displayed a decrease (23.2% vs. 31.0%) (Fig. 2I, J). According to a previous study[43], activated FAPs emerge in the early stage of muscle injury, therefore the phenomenon was in line with the delayed regeneration in dKO. Meanwhile, stressed FAPs were

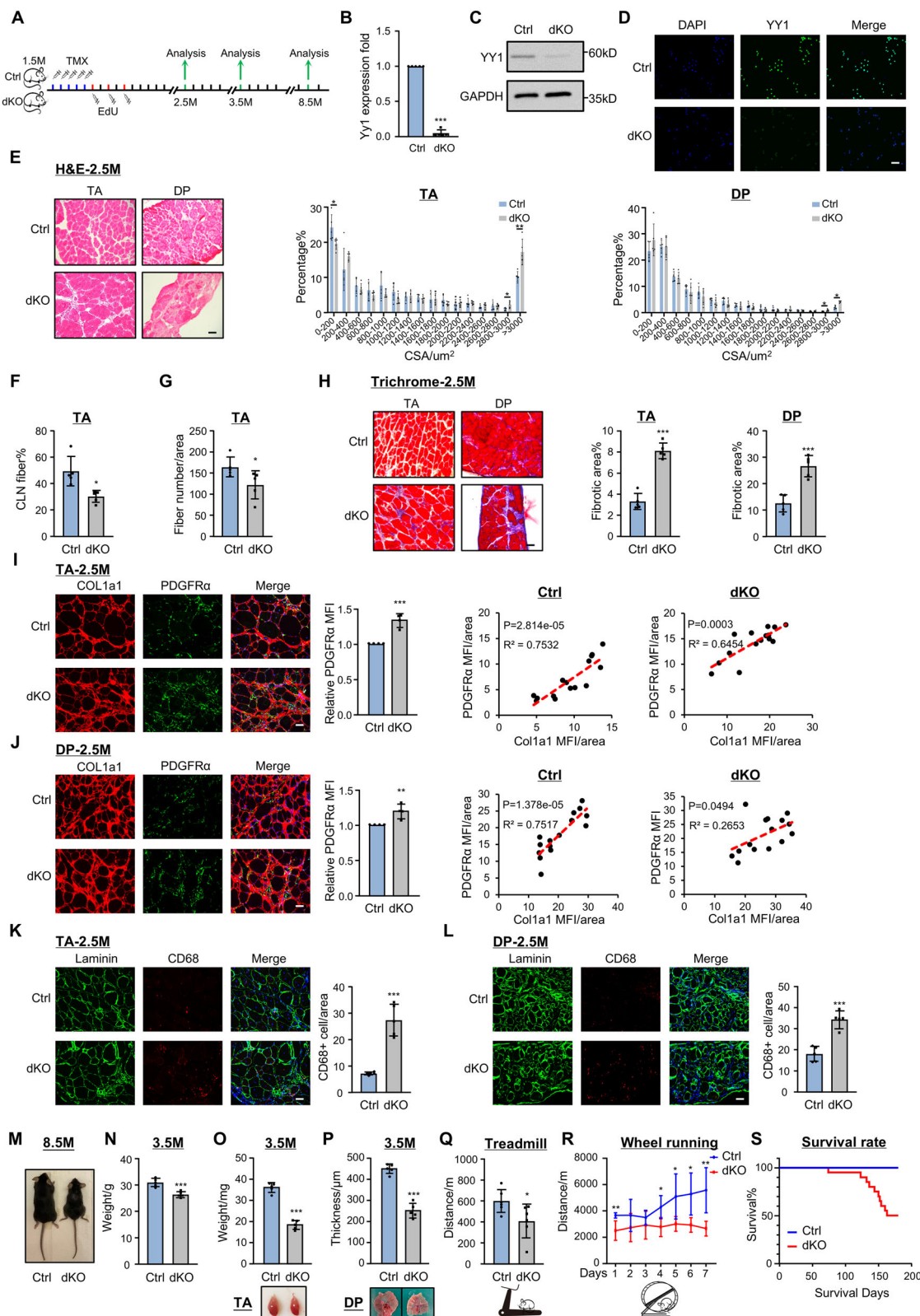

enriched with heat shock genes, which were closely related to cellular apoptosis[44,45], suggesting resistance to apoptosis in the dKO FAPs. Indeed, a significantly higher anti-apoptotic score (defined by anti-apoptotic gene set, Supplmentary Data 2) was measured in dKO vs. Ctrl FAPs (0.16 vs. 0.08) (Fig. 2K) alongside a decreased apoptotic score (Supplementary Fig. 2N, Supplementary Data 2). The above results suggest that enhanced apoptosis resistance may explain the elevated

FAP population in dKO muscle niche. It is also interesting to point out that an inflammatory FAP subset, marked by elevated expression of phagosome and chemokine pathway related genes, was identified exclusively in dKO but not in Ctrl group (Fig. 2I, J, Supplementary Fig. 2L, M).

When examining MO/MP subpopulations and heterogeneity, we identified five subsets in Ctrl muscle defined as 0-Monocyte, 1-

**Fig. 1 | Inducible deletion of YY1 in MuSCs aggravates muscle dystrophy in *mdx* mouse. A** Schematic of the experimental design for analyzing dystrophic phenotypes in Ctrl and dKO mice. Validation of YY1 ablation in dKO MuSCs by RT-qPCR (**B**), Western blot (**C**), and IF staining (**D**), $n = 5$ mice. $p = 0.0000014$ (**B**). **E** H&E staining of TA and DP muscles collected from the Ctrl and dKO mice at the age of 2.5 M. Distribution of fiber size is shown. Scale bar: 50 μm, $n = 5$ mice. $p = 0.034, 0.022, 0.044, 0.00015$. **F, G** Quantification of fibers with central-localized nuclei (CLN) and fiber numbers, $n = 5$ mice. $p = 0.0041$ (**F**), 0.0047 (**G**). **H** Masson's Trichrome staining on the above muscles. Quantification of fibrotic areas is shown. Scale bar: 50 μm, $n = 5$ mice. $p = 0.0000076, 0.00075$. **I, J** IF staining of Collagen1a1 (COL1a1, red) and PDGFRα (green) on the above muscles. Mean fluorescence intensity (MFI) and correlation of COL1a1 and PDGFRα MFI were calculated. Scale bar: 50 μm, $n = 4$ mice. $p = 0.028$ (**I**), 0.0055 (**J**). **K, L** IF staining of DAPI (blue), Laminin (green) and CD68 (red) on the above muscles. Quantification of the number of CD68+ cells per area. Scale bar: 50 μm, $n = 5$ mice. $p = 0.000074$ (**K**), 0.00013 (**L**). **M** Representative images of Ctrl and dKO mice at the age of 8.5 M. **N–P** Quantification of body weight, TA muscle weight and DP muscle thickness in the 3.5 M mice, $n = 5$ mice. $p = 0.00075$ (**N**), 0.0000014 (**O**), 0.0000051 (**P**). **Q** The running distance until exhaustion of the above mice subject to treadmill running, $n = 5$ mice. $p = 0.038$. **R** The daily running distance of the above mice subject to voluntary wheel-running, $n = 4$ mice. $p = 0.0094, 0.018, 0.033, 0.014, 0.0069$. Created in BioRender.com. **S** Survival rate of Ctrl and dKO mice before 6-month-old, $n = 10$ mice. All the bar graphs are presented as mean ± SD, paired two-sided Student's *t* test (**B, I, J**) and unpaired two-sided Student's *t* test (**E–H, K, L, N–R**) were used to calculate the statistical significance: *$p < 0.05$, **$p < 0.01$, ***$p < 0.001$, n.s. no significance. Source data are provided as a Source Data file.

Inflammatory, 2-Restorative, 3-MHC II, 4-Proliferating (Fig. 2L, Supplementary Fig. 2O, Supplementary Data 2)[43,46]. Monocytes were characterized by highly expressed *Cxcl3*, *Vcan* and *Chil3*; Inflammatory MPs were enriched for *Spp1*, *Fabp5* and *Cd36*; Restorative MPs expressed high levels of *C1qa*, *C1qb*, and *C1qc*; MHC II MPs harbored an abundance of *H2-Aa*, *H2-Eb1*, and *H2-Ab1*; The Proliferating subset was distinguished by highly expressed cell cycle and cell division related genes such as *Ccna2*, *Ccnb2*, *Cdk1*, *Cdc20*, *Cdca3*, *Cdca8* (Supplementary Fig. 2O). In dKO, Monocyte, Inflammatory, Restorative, MHC II but not Proliferating subsets were identified; interestingly, a unique subpopulation with highly induced CCL family genes (*Ccl7*, *Ccl4*, *Ccl2*, *Ccl8*, *Ccl6*, *Ccl24*, *Ccl3*) was detected and named CCL+ MPs (Fig. 2M and Supplementary Fig. 2P). When the pseudotime trajectory was analyzed, we found that in Ctrl (Fig. 2L) Monocytes plotted tightly together at the initial position of the trajectory line, followed by bifurcation into Restorative and Proliferating MPs. Inflammatory and MHC II MPs were distributed along the line between Monocyte and Restorative MPs. A distinct trajectory was plotted in dKO (Fig. 2M): starting from Monocyte, the line diverged into three branches, Restorative, MHC II, CCL+; Inflammatory mainly located along the two lines between Monocyte and Restorative/CCL+. The proportions of both Inflammatory and Restorative were increased in dKO vs. Ctrl (23.6% vs. 22.3%, 38.2% vs. 26.4%), but the MHC II MPs showed a decline (14.3% vs. 19.2%) (Fig. 2L, M); the dKO specific CCL+ cells displayed enhanced inflammatory features (enriched for *Rsad2*, *Ifit1* and *Tnf*) while the Ctrl specific Proliferating subset exhibited neutrophil-like scavenger characteristics (enriched for *S100a8*, *Camp* and *Ngp*). Altogether, the above results support that the inflammatory niche is skewed in dKO muscle. Consequently, by comparing the inflammatory score (defined by inflammatory gene set, Supplementary Data 2), a significant elevation was detected (0.16 vs. −0.08) (Fig. 2N). Altogether, scRNA-seq results show that the intrinsic deletion of YY1 in MuSCs leads to niche remodeling primarily by altered FAP and MP compositions.

## Intrinsic deletion of YY1 in MuSCs enhances their crosstalk with MPs via CCL5/CCR5 axis

To delineate the molecular mechanism underlying the cellular composition changes in dKO muscles, bulk RNA-seq was performed on freshly isolated MuSCs from six pairs of Ctrl or dKO mice (Fig. 3A, Supplementary Data 3). A total of 1090 genes were up-regulated and 1527 down-regulated in dKO compared with Ctrl (Fig. 3B and Supplementary Data 3). Strikingly, Gene Ontology (GO) analysis revealed that the up-regulated genes were highly enriched for immune related terms such as "immune system process", "innate immune response" etc. (Fig. 3C and Supplementary Data 3), suggesting an immune-like nature of dKO cells. Conversely, genes associated with "cell cycle" and "cell division" were predominantly down-regulated (Fig. 3D and Supplementary Data 3), aligning with the detected reduced proliferative capacity of dKO MuSCs (Supplementary Fig. 1D, E and Supplementary Data 3). A detailed examination of the up-regulated genes highlighted the induction of numerous pro-inflammatory mediators, particularly those facilitating MPs infiltration into injured tissue, such as a panel of chemoattractants from CCL family, *Ccl5*, *Ccl25*, *Ccl2*, *Ccl7*, *Ccl3* (Supplementary Fig. 3A, B and Supplementary Data 3)[26,47,48]. Among these factors, *Ccl5* gene induction was robustly detected in dKO MuSCs (Fig. 3E, 2.58-fold, Supplementary Fig. 3C–E) accompanied by the protein induction (Fig. 3F, 72.89% vs. 3.55%) and secretion (Fig. 3G); a gradually increased amount of CCL5 protein was also detected in dKO muscles after TMX injection (Fig. 3H). It is known that enriched ligands can promote the up-regulated receptor expression in cells[49]; expectedly, we found an escalated amount of *Ccl5* receptor *Ccr5* mRNA and protein in dKO muscles (Fig. 3I). Moreover, co-localization of CCL5 and CCR5 proteins were detected (Fig. 3J). CCL5/CCR5 axis is known to play an important role in chemoattracting MPs[50–52], consistently, we found *Ccr5* was mainly expressed by MPs (Supplementary Fig. 3F) and a significantly increased expression was detected in dKO MPs (Supplementary Fig. 3G) by analyzing the scRNA-seq data, which was further confirmed in the isolated MPs from TMX-1M muscle (Fig. 3K). Moreover, IF staining showed that most of the Ctrl and dKO MPs were CCR5+ cells (>95%, Supplementary Fig. 3H). Altogether, these findings suggest that CCL5 induction in dKO MuSCs enhances MP recruitment via a CCL5/CCR5 axis. To further test this, transwell assay was performed by seeding MuSCs from Ctrl or dKO underneath the transwell insert and bone marrow derived macrophages (BMDMs, Supplementary Fig. 3I) on the top (Fig. 3L). As expected, a much higher number of migrated BMDMs were detected in dKO vs. Ctrl (113.3 vs. 79.4, Fig. 3L), indeed confirming the enhanced recruiting ability of dKO MuSCs. To validate the role of CCL5/CCR5 axis in mediating the recruitment, we found that BMDM attraction by dKO cells was significantly attenuated by neutralization of CCL5 (Fig. 3M) or CCR5 (Fig. 3N) with antibodies. Altogether, the above findings demonstrate that intrinsic loss of YY1 transforms MuSCs into immune-secretory and endows its heightened ability to crosstalk and recruit MPs to the niche microenvironment via the CCL5/CCR5 axis.

## TGFβ1 enrichment in the niche causes FAP accumulation in dKO muscle by inhibiting apoptosis

To further elucidate the underlying cause for increased FAPs in dKO muscle, we examined if the dKO FAPs possess apoptosis resistance as suggested by Fig. 2K. PDGFRα + FAPs were isolated from Ctrl and dKO mice and used for TUNEL staining (Fig. 4A). As expected, a significant reduction (32.5%) of TUNEL+ cells was observed in dKO vs. Ctrl (Fig. 4B), confirming the resistance to apoptosis of dKO FAPs. This was further supported by co-staining of TUNEL and PDGFRα on TA (Fig. 4C) or DP (Fig. 4D) muscles, showing a substantial decrease (26.3%, 9.3%) of apoptotic FAPs in dKO. We also examined the proliferative ability of FAPs by EdU staining (Fig. 4A); no significant difference of EdU+ cells was detected when the assays were conducted on in vitro cultured (Fig. 4E, Supplementary Fig. 4A, B) and in vivo isolated (Fig. 4F, Supplementary Fig. 4C, D) FAPs from Ctrl or dKO mice. Moreover, the fibrogenic and adipogenic differentiation abilities of FAPs were examined. Elevated α-SMA protein (Supplementary Fig. 4E)

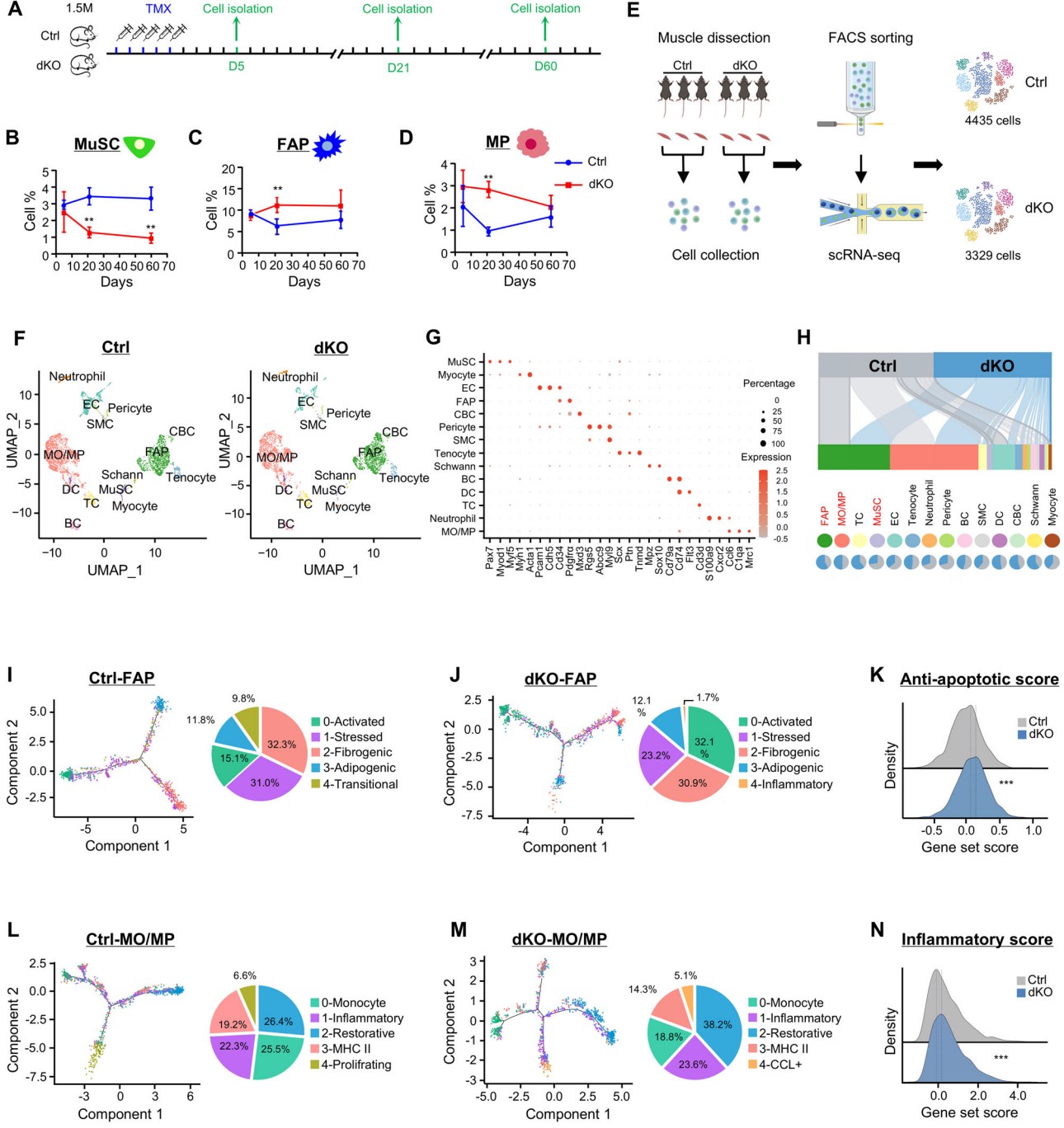

**Fig. 2 | Intrinsic deletion of YY1 in MuSCs alters cellular microenvironment in dystrophic muscle. A** Schematic of the experimental design for analyzing MuSC, FAP, and MP populations in Ctrl and dKO mice. **B–D** The percentages of isolated MuSCs, FAPs, and MPs from Ctrl and dKO mice at 5, 21 and 60 days after TMX administration, $n = 9$ mice. $p = 0.0034$, 0.0055 (**B**); 0.0085 (**C**); 0.0013 (**D**). **E** Schematic of scRNA-seq experimental design (created in BioRender.com). Mononuclear cells from three pairs of Ctrl and dKO mice were combined for FACS sorting and living cell selection. A total of 4435 and 3329 living cells from Ctrl and dKO mice were identified respectively. **F** scRNA-seq was performed in the whole muscle from three pairs of Ctrl and dKO mice. Uniform manifold approximation and projection (UMAP) is shown to visualize variation in single-cell transcriptomes. Unsupervised clustering resolved at least 14 cell types (color coded). **G** Dot plot showing the expression signatures of representative marker genes for each cell type. **H** Top: Sankey plots showing the distribution of Ctrl and dKO cells across different cell types. Bottom: pie plots showing the relative cell proportion between Ctrl and dKO groups across different cell types. **I, J** Left: pseudotime trajectory inference of the identified FAP subpopulations in Ctrl and dKO. Right: pie charts showing the relative cell proportion of each subtype. **K** Ridge map showing the global distribution density of anti-apoptotic of Ctrl and dKO FAPs. The corresponding dashed line represents the peak position of each group. $p = 0.0000028$. **L, M** Left: pseudotime trajectory inference of the identified MO/MP subpopulations in Ctrl and dKO. Right: pie charts showing the relative cell proportion of each subtype. **N** Ridge map showing the global distribution density of inflammatory scores of Ctrl and dKO MO/MP. The corresponding dashed line represents the peak position of each group. $p = 0.00000000000000022$. All the bar graphs are presented as mean ± SD, unpaired two-sided Student's $t$ test was used to calculate the statistical significance (**B–D**, **K**, **N**): *$p < 0.05$, **$p < 0.01$, ***$p < 0.001$, n.s. no significance. Source data are provided as a Source Data file.

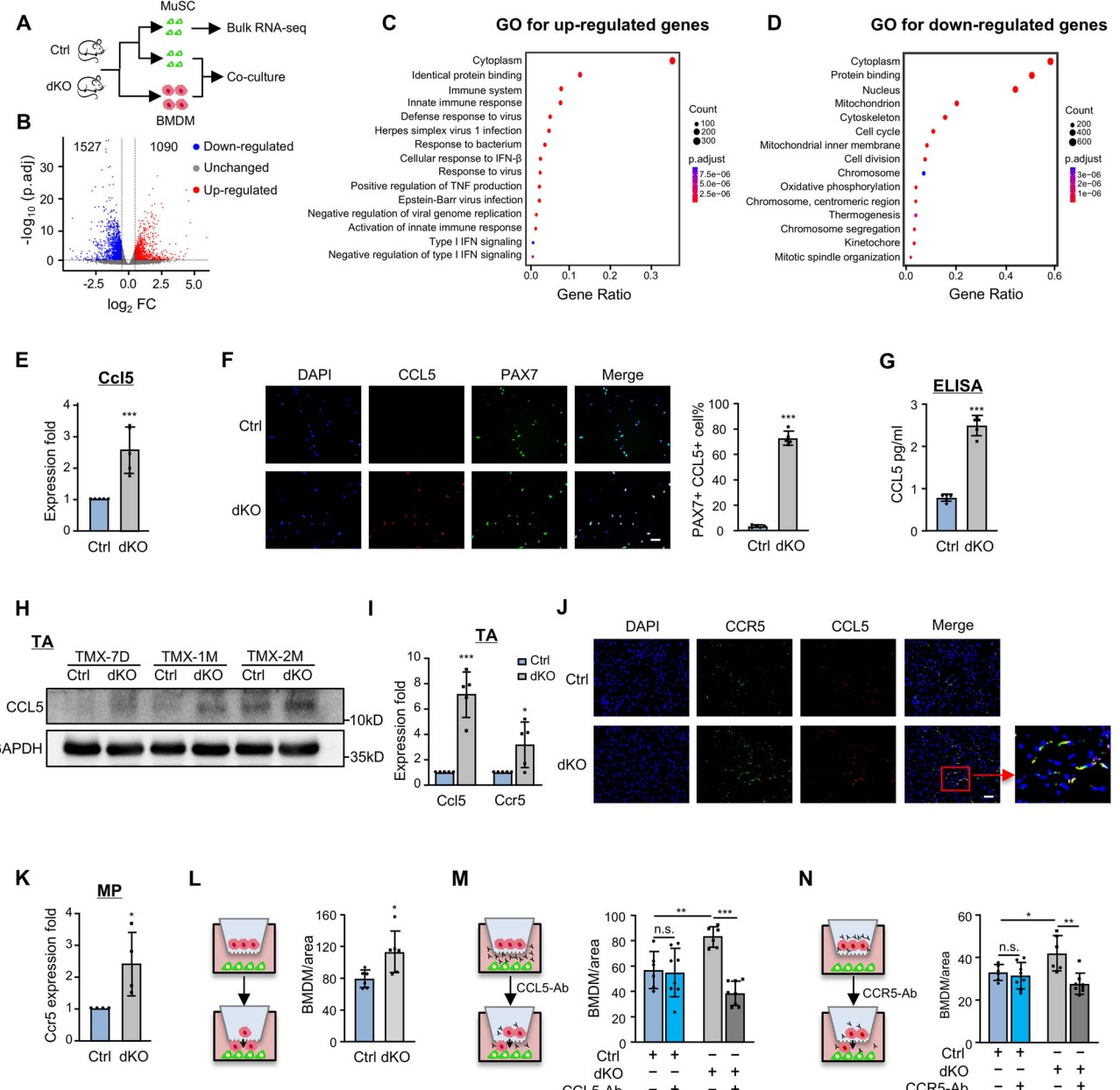

**Fig. 3 | Intrinsic deletion of YY1 in MuSC induces enhanced crosstalk between MuSC and MP via CCL5/CCR5 axis. A** Schematic of the experimental design for testing MuSC/MP interaction in Ctrl and dKO mice. **B** Differentiability expressed genes (DEGs) were identified from the RNA-seq profiling in Ctrl vs dKO MuSCs using Log$_2$FC > 0.5 as a cut-off. **C, D** GO analysis of the above identified 1090 up and 1527 down-regulated DEGs. **E** RT-qPCR detection of *Ccl5* mRNA in freshly isolated MuSCs form Ctrl and dKO, $n = 5$ mice. $p = 0.0086$. **F** IF staining of CCL5 protein in freshly isolated MuSCs from Ctrl and dKO. Scale bar: 50 μm, $n = 5$ mice. $p = 0.0000000031$. **G** ELISA detection of secreted CCL5 protein from Ctrl and dKO MuSCs, $n = 5$ mice. $p = 0.00000036$. **H** Western blot detection of CCL5 protein in TA muscles of Ctrl and dKO at the designated times after TMX administration. **I** RT-qPCR detection of *Ccl5* and *Ccr5* mRNAs in Ctrl and dKO TA muscles, $n = 5$ mice. $p = 0.0015, 0.0498$. **J** IF staining of CCL5 and CCR5 proteins on TA muscle sections

of Ctrl and dKO. Co-localization of CCL5 and CCR5 is shown in the red frame. Scale bar: 50 μm, $n = 5$ mice. **K** RT-qPCR detection of *Ccr5* mRNA in MPs freshly isolated from Ctrl and dKO, $n = 4$ mice. $p = 0.032$. **L** BMDMs were isolated from *mdx* mice and co-cultured with MuSCs from Ctrl and dKO in transwell. Quantification of migrated BMDMs is shown, $n = 6$ mice. $p = 0.015$. **M, N** CCL5 or CCR5 antibody was added to the above transwell. Quantification of migrated BMDMs is shown, $n = 6$ mice. $p = 0.0026, 0.00000073$ (**M**); $0.035, 0.0016$ (**N**). All the bar graphs are presented as mean ± SD, unpaired one-sided Student's *t* test (**B**), paired two-sided Student's *t* test (**E, I, K**), unpaired two-sided Student's *t* test (**F, G, L–N**) and one-sided Fisher's exact test (**C, D**) were used to calculate the statistical significance: *$p < 0.05$, **$p < 0.01$, ***$p < 0.001$, n.s. no significance. Source data are provided as a Source Data file.

and mRNA (Supplementary Fig. 1F) were detected in dKO TA muscle, indicating increased fibrogenic differentiation propensity. However, adipogenic assessment by Oil red O staining or marker gene expressions showed no significant difference (Supplementary Fig. 4F, G). Altogether, these results demonstrate that the accumulation of FAPs in dKO muscle mainly arises from the mitigated apoptosis thus the

enhanced survival and the increased fibrogenic differentiation may exacerbate the fibrosis. To assess if direct crosstalk from MuSCs impacts FAP apoptosis and proliferation in dKO muscle niche (Fig. 4A), MuSCs from Ctrl or dKO mice were co-cultured with FAPs (isolated from Ctrl mice) for 24 hours (Fig. 4G) and there were no significant changes in apoptosis by TUNEL staining (Fig. 4H) or proliferation by

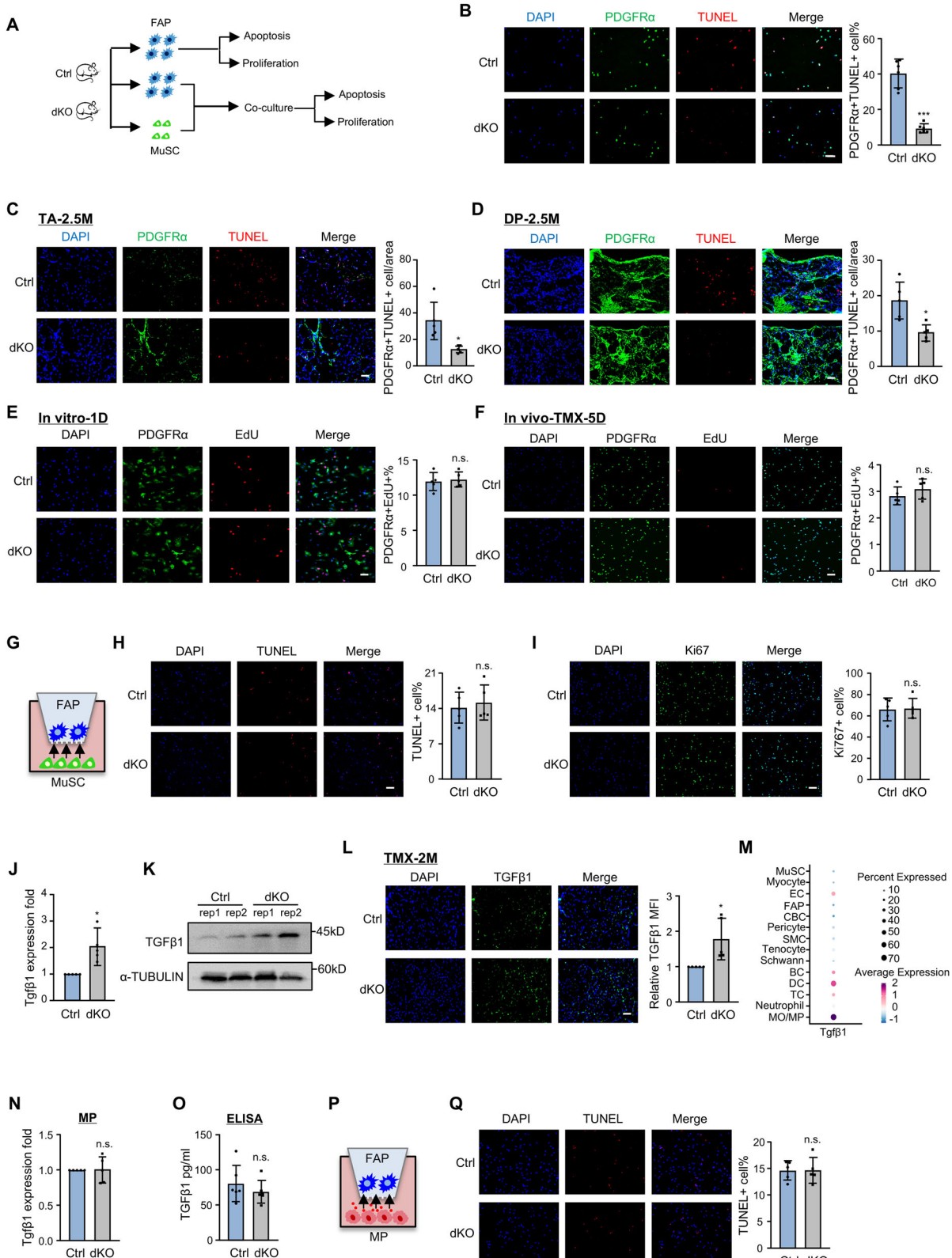

Ki67 staining (Fig. 4I), suggesting that MuSCs may exert negligible direct impact on FAPs.

We then sought to test the possibility that enhanced FAP survival in dKO muscle may be caused by increased TGFβ1 level since it was recently demonstrated that highly expressed TGFβ1 prevented FAP apoptosis and promoted their differentiation into matrix-producing cells, contributing to fibrosis in dystrophic muscle[29]. No obvious

difference in TGFβ1 expression was detected in dKO vs. Ctrl muscles at TMX-7D (Supplementary Fig. 4H), but a significant induction was observed at later stages (TMX-14D, Supplementary Fig. 4I; TMX-2M, Fig. 4J–L), indicating the gradual accumulation of TGFβ1 in the muscle niche after YY1 deletion which is concomitant with FAP increase. According to previous reports[29,40,53], the predominant contributor of TGFβ1 in dystrophic muscle is MPs, indeed, analyzing our scRNA-seq

**Fig. 4 | TGFβ1 enriched niche inhibits FAP apoptosis and causes FAP accumulation in dKO muscle. A** Schematic of the experimental design for testing FAP/MuSC interaction in Ctrl and dKO mice. **B** TUNEL staining of FAPs isolated from Ctrl and dKO muscles. The percentage of TUNEL+ cells is shown. Scale bar: 50 μm, *n* = 6 mice. *p* = 0.0000041. **C, D** TUNEL and PDGFRα staining of TA or DP muscles from Ctrl and dKO mice. The percentage of TUNEL + PDGFRα+ cells is shown. Scale bar: 50 μm, *n* = 5 mice. *p* = 0.0084 (**C**), 0.0095 (**D**). **E** EdU staining of 1D-in vitro cultured FAPs. The percentage of EdU+ cells is shown. Scale bar: 50 μm, *n* = 5 mice. *p* = 0.70. **F** EdU and PDGFRα staining of freshly isolated FAPs from TMX-5D mice. The percentage of EdU+ PDGFRα+ cells is shown. Scale bar: 50 μm, *n* = 5 mice. *p* = 0.28. **G** Schematic of MuSCs and FAPs co-culture experiment. **H** TUNEL staining of above co-cultured FAPs. The percentage of TUNEL+ cells is shown. Scale bar: 100 μm, *n* = 5 mice. *p* = 0.64. **I** Ki67 staining of the above co-cultured FAPs. The percentage of Ki67+ cells is shown. Scale bar: 100 μm, *n* = 5 mice. *p* = 0.088. **J, K** RT-qPCR and Western blot detection of TGFβ1 mRNA and protein in TA muscles, *n* = 5 mice. *p* = 0.030. **L** IF staining of TGFβ1 and quantification of MFI in TMX-2M TA muscles. Scale bar: 50 μm, *n* = 5 mice. *p* = 0.041. **M** Bar plot showing the *Tgfβ1* expression in cells. **N** RT-qPCR detection of *Tgfβ1* expression in Ctrl and dKO MPs, *n* = 5 mice. *p* = 0.996. **O** ELISA detection of secreted TGFβ1 protein from Ctrl and dKO MPs, *n* = 6 mice. *p* = 0.38. **P** Schematic of MPs and FAPs co-culture experiment. **Q** TUNEL staining of above co-cultured FAPs. The percentage of TUNEL+ cells is shown. Scale bar: 50 μm, *n* = 5 mice. *p* = 0.996. All the bar graphs are presented as mean ± SD, paired two-sided Student's *t* test (**J, L, N**) and unpaired two-sided Student's *t* test (**B–F, H, I, O, Q**) were used to calculate the statistical significance: *p < 0.05, **p < 0.01, ***p < 0.001, n.s. no significance. Source data are provided as a Source Data file.

data uncovered MPs as the main source of TGFβ1 (Fig. 4M). To further investigate whether the elevated TGFβ1 is a result of increased level of secretion or number of MPs, we found no difference in TGFβ1 expression (Fig. 4N) or secretion (Fig. 4O) when the same number of MPs from Ctrl or dKO were analyzed. To further test the impact of MP secreted TGFβ1 on FAP apoptosis, the same number of MPs from Ctrl or dKO mice were co-cultured with FAPs (isolated from Ctrl) for 24 h (Fig. 4P) and no significant change in apoptosis was detected by TUNEL staining (Fig. 4Q); but when a higher ratio of dKO cells (5:3, dKO: Ctrl, mimicking the in vivo situation) were used in the assay, FAPs displayed significantly decreased apoptosis (Supplementary Fig. 4J), suggesting that the enriched TGFβ1 accumulation in dKO muscle is more likely a direct result of the increased number of MPs. Altogether, these findings demonstrate that the heightened MP population results in elevated TGFβ1 level, which inhibits FAP apoptosis thus causes FAP accumulation to exacerbate fibrosis in dKO muscle.

## Inhibiting CCL5/CCR5 signaling axis with Maraviroc alleviates muscle dystrophy

Next, to investigate if inhibiting above-defined CCL5/CCR5 signaling can mitigate dystrophy in dKO mice, we treated Ctrl and dKO mice with MVC, a well-documented CCR5 antagonist[54,55]. IP administration of MVC in Ctrl and dKO mice was repeated every other day over 30 days at 2 mg/kg after which the TA muscles were isolated for subsequent analysis (Fig. 5A). The treatment did not affect the expression of CCL5 in MuSCs (Supplementary Fig. 5A, B) or TA muscles (Supplementary Fig. 5C), but significantly ameliorated the muscle phenotypes in dKO mice: the regeneration was elevated, evidenced by decreased abnormally larger fibers (Fig. 5B), increased fibers with CLN (Fig. 5C) and fiber numbers (Fig. 5D), accompanied by significantly tampered amount of fibrosis (Fig. 5E). As expected, a concomitant decline of the numbers of FAPs and MPs was observed in the dKO muscle niche (Fig. 5F–H), accompanied by decreased TGFβ1 expression (Supplementary Fig. 5D, E), but the number of MuSCs did not show any obvious increase (Fig. 5F).

Consistently, the treated mice displayed significantly improved muscle function when subject to treadmill or voluntary wheel-running exercises. The endurance of dKO mice was notably recovered in the treadmill experiment (Fig. 5I). The treated dKO mice also demonstrated increased engagement in voluntary wheel-running, with a consistent increase in running distance over time (Fig. 5J). The overall morphology and health state of the dKO mice were improved by the treatment showing a significantly restored body weight (Fig. 5K). Altogether, these results further underpin the importance of CCL5/CCR5 signaling axis in muscle dystrophy and highlight the possibility of targeting this axis to modulate muscle niche and decelerate disease progression.

## YY1 controls *Ccl5* expression in MuSC via regulating 3D E-P loop interaction

To answer how intrinsic YY1 deletion in MuSCs induces *Ccl5* expression, YY1 ChIP-seq was employed to map YY1 bound genes in freshly isolated MuSCs from *mdx* mice (Fig. 6A). A total of 4681 YY1 binding sites were identified with a canonical YY1 binding motif "AANATGGC" (Fig. 6B and Supplementary Data 4). Over half of the binding occurred in promoter regions (59%) while 29% and 12% in gene body and intergenic regions (Fig. 6C). GO analysis revealed YY1 bound genes were enriched for a wide range of terms such as "histone modification" and "mRNA processing" etc. (Fig. 6D). Further integration with the RNA-seq identified DEGs (Fig. 3B) uncovered a total of 181 and 226 YY1 bound genes were up- and down-regulated in dKO vs. Ctrl MuSCs (Fig. 6E and Supplementary Data 4). GO analysis revealed that the up-regulated targets were highly enriched for "Nucleus", "Nucleoplasm" and "DNA binding" etc. while the down-regulated targets were related to "Nucleus", "Nucleoplasm" and "Cell cycle" etc. (Supplementary Fig. 4A, B and Supplementary Data 4). Next, we took a close examination of the *Ccl5* locus and found no YY1 binding at its promoter region, instead, a binding site was identified at a distal site which was a potential enhancer by analyzing our previously published H3K27ac ChIP-seq data[56] (Fig. 6F). Considering YY1 is emerging as a 3D chromatin organizer that can facilitate enhancer-promoter (E-P) looping[57–59], we tested if YY1 modulated *Ccl5* transcription via orchestrating E-P looping in MuSCs. To this end, we performed in situ Hi-C[60] to interrogate the 3D genome organization in Ctrl and dKO MuSCs. We found that YY1 knock-out in MuSCs had limited effect on global 3D genome organization as no difference in the contact frequency was detected between Ctrl and dKO at both short- and long-distance levels (Supplementary Fig. 4C). The YY1 knock-out did not alter the overall genome organization at compartment level either (Supplementary Fig. 4D), indicated by infrequent locus switching between A and B compartments, with only 2.8% B to A and 2.2% A to B switching (Fig. 6G). At the TAD level, however, we noticed an evident TAD boundary remodeling: 9393 new boundaries formed while 4033 out of 13416 remained unchanged (Fig. 6H); and a significant decline in the TAD size was detected in dKO vs. Ctrl (Fig. 6I). Additionally, a slight increase in the number of TADs was observed (9557 in dKO vs. 8965 in Ctrl) (Fig. 6J). Furthermore, at the looping level, we found the loop size remained largely unaltered (Fig. 6K) while a declined number of looping events occurred in dKO vs. Ctrl (1056 vs. 1753) (Fig. 6L). These results suggest that YY1 may play a role in regulating the 3D genome in MuSCs via looping modulation.

Next, we took a close examination of the *Ccl5* locus, attempting to elucidate the cause behind high *Ccl5* in dKO MuSCs. Interestingly, two evident TADs were observed encompassing the *Ccl5* locus in Ctrl; and a newly gained third one was found in dKO and the above-identified YY1 bound enhancer was located within this gained TAD (Fig. 6M). Further analysis of E-P looping defined one strong looping event occurring between this enhancer and the *Ccl5* promoter; moreover, the interacting strength measured by interaction frequency was significantly lower in dKO vs. Ctrl (7.6 vs. 16.5) (Fig. 6M), suggesting an interesting scenario where YY1 mediated E-P looping functions to suppress *Ccl5* induction in Ctrl and the attenuation of the looping relieves the suppression in dKO. To test this notion, the above-identified E-P

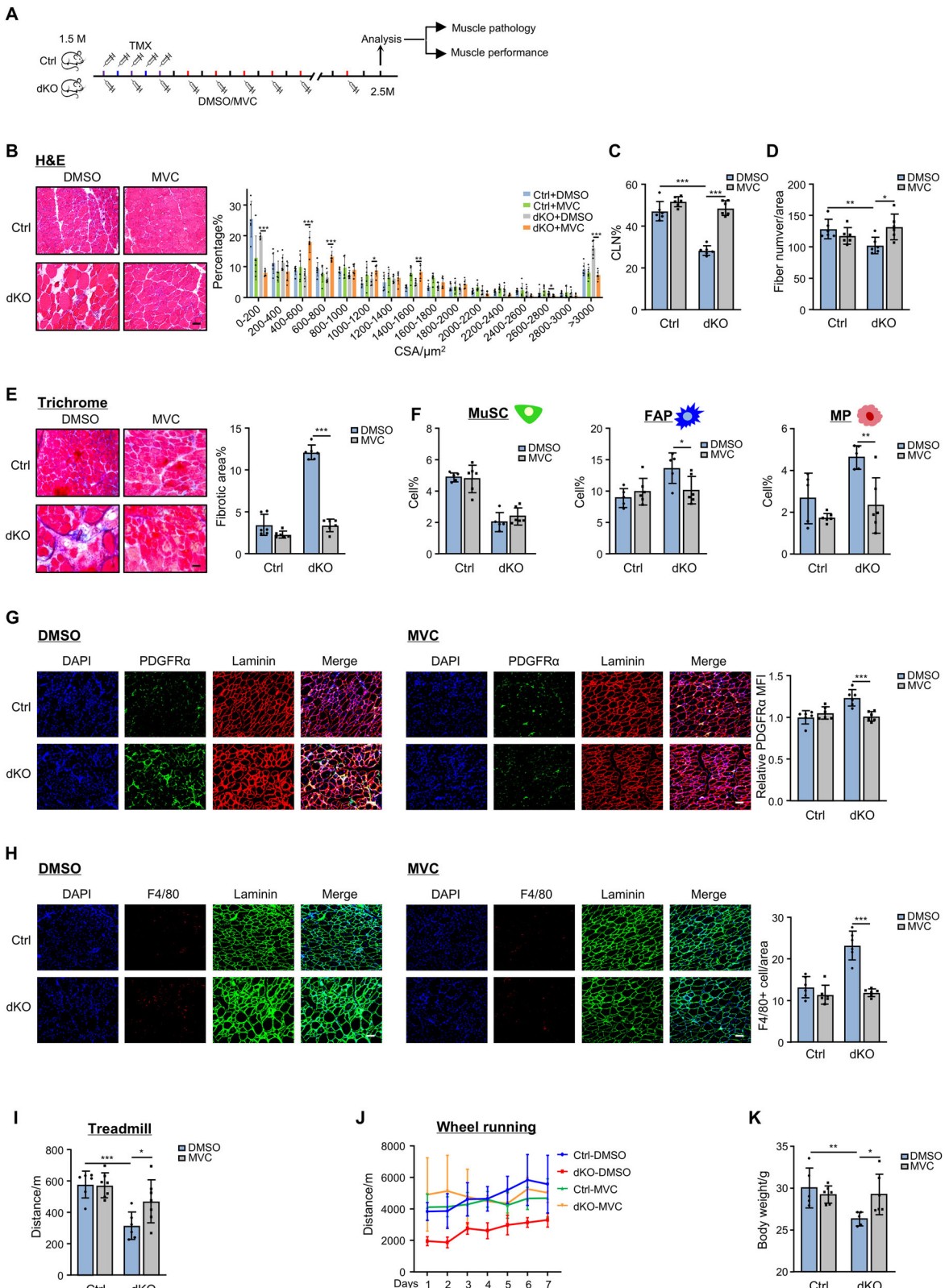

interaction was validated by Chromosome Conformation Capture (3C) assay in freshly isolated MuSCs from Ctrl and dKO using the *Ccl5* promoter as an anchor (Fig. 6N). Expectedly, a prominent interaction peak was detected at the enhancer region in both Ctrl and dKO groups and the quantified interaction frequency by 3C-qPCR weakened as the genomic distance from the enhancer site increased. In addition, the interaction frequency at the enhancer site was reduced in dKO vs. Ctrl

(0.16 vs. 0.10) (Fig. 6O). These results indicate that the YY1 deletion in MuSCs indeed attenuates the E-P interaction at the *Ccl5* locus.

To further substantiate the direct function of YY1 in facilitating the E-P interaction, artificial tethering of YY1 protein to the *Ccl5* enhancer region (Fig. 6P) was performed in C2C12 myoblast cell line where no direct binding of YY1 on the *Ccl5* enhancer locus was detected by analyzing the existing H3K27ac and YY1 ChIP-seq data

**Fig. 5 | Targeting CCL5/CCR5 axis with MVC alleviates muscle dystrophy.**
**A** Schematic of the DMSO or MVC treatment and assessment in six pairs of male Ctrl and dKO mice. **B** H&E staining of TA and DP muscles collected from the Ctrl and dKO mice at the age of 2.5 M. Distribution of fiber size is shown. Scale bar: 50 μm, $n = 6$ mice. $p = 0.00000000061, 0.000051, 0.000061, 0.032, 0.0048, 0.000016$. **C** Quantification of CLN+ fibers in the above-stained muscles, $n = 6$ mice. $p = 0.0000049, 0.00000036$. **D** Quantification of fiber numbers per area of the above-stained muscles, $n = 6$ mice. $p = 0.0096, 0.013$. **E** Masson's Trichrome staining on the above muscles. Quantification of fibrotic areas is shown. Scale bar: 50 μm, $n = 6$ mice. $p = 0.0000000044$. **F** Flow cytometry detection of MuSC, FAP, and MP population in Ctrl and dKO mice after the treatment, $n = 5$ mice. $p = 0.038, 0.0056$. **G** IF staining of DAPI (blue), PDGFRα (green), and Laminin (red) was

performed on TA muscles of Ctrl and dKO mice after the treatment. The quantification of MFI of PDGFRα is shown. Scale bar: 50 μm, $n = 6$ mice. $p = 0.00078$. **H** IF staining of DAPI (blue), F4/80 (red), and Laminin (green) was performed on TA muscles of Ctrl and dKO mice after the treatment. The quantification of F4/80+ cell number per area is shown. Scale bar: 50 μm, $n = 6$ mice. $p = 0.000015$. **I** The running distance until exhaustion of the above mice subject to treadmill running is shown, $n = 6$ mice. $p = 0.00037, 0.037$. **J** The daily running distance of the above mice subject to voluntary wheel-running is shown, $n = 6$ mice. **K** Body weight of Ctrl and dKO mice after the treatment, $n = 5$ mice. $p = 0.0010, 0.031$. All the bar graphs are presented as mean ± SD, unpaired two-sided Student's $t$ test was used to calculate the statistical significance (**B–I**, **K**): $*p < 0.05$, $**p < 0.01$, $***p < 0.001$, n.s. no significance. Source data are provided as a Source Data file.

from C2C12 myoblast[16,61] (Supplementary Fig. 4E). A dCas9-YY1 plasmid and five sgRNAs targeting the enhancer region were harnessed to generate five stable clones expressing increased level of dCas9-YY1; the transfection efficiency was confirmed in both mRNA (Supplementary Fig. 4F) and protein levels, with no significant influence on endogenous YY1 expression (Supplementary Fig. 4G). As a result, YY1 enrichment at the *Ccl5* enhancer region was markedly strengthened after tethering (Fig. 6Q) (detected by YY1 ChIP-qPCR using five primers targeting the enhancer region) and an increased contact frequency (0.0036 vs. 0.0012) was detected between the promoter and the enhancer by 3C-qPCR (Fig. 6R). Accordingly, the expression level of *Ccl5* was significantly reduced in four of the five stable clones (Fig. 6S). Altogether, the above results demonstrate an active role of YY1 in facilitating E-P interaction to curb the activation of *Ccl5* locus in Ctrl MuSCs. The regulatory role of YY1 and the upstream enhancer was further confirmed by transfecting a luciferase reporter containing the promoter and enhancer regions of *Ccl5* in Ctrl and dKO MuSCs (Fig. 6T). A significantly higher level of luciferase activity was detected in dKO vs. Ctrl (8.73 vs. 1.00) (Fig. 6T). Moreover, a reporter devoid of the enhancer sequence displayed elevated reporter activity (20.86 vs. 1.00) in Ctrl MuSCs (Fig. 6U), solidifying the repressive function of the enhancer element. Altogether, these results have confirmed the direct role of YY1 in repressing the *Ccl5* transcription by enhancing the E-P interaction.

## Discussion

In this study, we leverage the *Yy1/mdx* dKO mouse strain to elucidate how MuSCs impact their niche microenvironment in dystrophic muscle. We demonstrate the MuSCs are capable of modulating their niche via cellular interactions with MPs and FAPs. Intrinsic deletion of YY1 in MuSCs leads to unexpected aggravation of inflammation and fibrosis in dKO which is attributed to altered niche microenvironment characterized by skewed dynamics and heterogeneity of MPs and FAPs. Further investigation demonstrates that conveying the dKO's intrinsic change to niche alterations stems from the induction of many immune related genes causing the conversion of MuSCs into competent secretory cells. In particular, we identify a CCL5/CCR5 signaling axis that mediates the MuSC-MP cellular interaction which enhances MP recruitment and exacerbates inflammation. As a result, elevated TGFβ1 secreted by MPs leads to FAP accumulation and aggravated fibrosis. Treatment with the CCR5 antagonist MVC reduces dystrophic disease manifestations and improves muscle function. Furthermore, we also illuminate the intrinsic mechanism of how YY1 loss causes *Ccl5* induction in MuSCs via YY1's ability to modulate enhancer-promoter looping interaction at *Ccl5* locus (Fig. 7).

Our study provides compelling evidence to demonstrate the important active role of MuSCs in modulating niche microenvironment. As the small population of residents in skeletal muscle, MuSCs are often portrayed as the passive recipient of niche regulation. Mounting efforts are focused on dissecting how MuSC behavior is tightly regulated by spatiotemporal signaling from the niche and how niche-derived growth factors and signaling molecules, metabolic cues,

the extracellular matrix and biomechanical cues, and immune signals exert their effects on MuSCs[62]. However, sporadic evidence hinted the possible reciprocal effect that MuSCs exert on MPs. For example, a study from 2013 demonstrated that human myoblasts can indeed secrete an array of chemotactic factors to initiate monocyte chemotaxis in culture[26]. A recent study from Oprescu et al.[43] profiled transcriptomic dynamics at various stages of acute damage-induced muscle regeneration by scRNA-seq; a myoblast subpopulation enriched for immune genes were identified to be capable of active communication with immune cell populations. More recently, a study from Nakka K. et. al. demonstrates MuSCs can initiate the production of hyaluronic acid to modulate the ECM in the niche which in turn directs MuSCs to exist quiescence state for repair of injured muscle[63]. Moreover, emerging reports reveal the presence of senescent MuSCs in regenerating and aging muscles; these senescent MuSCs are characterized by SASPs (senescence-associated secretory phenotypes) and could have paramount roles in remodeling niche microenvironment[64]. Therefore, following our study, much effort will be needed to elucidate the functional interactions between MuSCs and the niche. Moreover, we also posit that similar niche-regulatory functions might be a feature of other adult stem cells; the tremendous advances in single cell profiling and spatial transcriptomic technologies are poised to revolutionize our understanding of stem cell niche dynamics in unparalleled detail.

Our findings uncovered a novel MuSC/MP interaction that offers insights into the pathogenic recruitment of MPs in dystrophic muscle. In acutely injured skeletal muscle, infiltrating MPs are mainly Ly6C$^{hi}$ CCR2$^+$CX3CR1$^{lo}$ monocytes that are recruited through CCR2 chemotaxis signaling by myoblasts, ECs and resident macrophages[36]. These monocytes then differentiate into Ly6C$^{hi}$ inflammatory MPs. After phagocytosing necrotic muscle debris, intramuscular Ly6C$^{hi}$ MPs can switch into Ly6C$^{lo}$ MPs. It has been shown that recruitment of Ly6$^{hi}$ monocytes in *mdx* mouse muscles is also dependent on CCR2[51]. Nevertheless, the induction of other CCL class chemokine ligands including CCL5, CCL6, CCL7, CCL8, and CCL9 and receptors CCR1, and CCR5, has also been observed in *mdx* mouse muscle[65]. Here we demonstrate that MuSCs can also contribute to MP recruitment via the CCL5/CCR5 axis. MuSC-MP interactions may have commenced promptly following TMX injection, evidenced by immediately elevated expression and secretion of CCL5 in MuSCs and an increased population of MPs in dKO. Nevertheless, MuSCs may not be the only source of CCL5, which can also be a product of T cells, MPs and muscle fibers[65]. Additionally, it will be interesting to further explore whether the MuSC-MP crosstalk has impact on the heterogeneous presence of MPs in dystrophic muscles. It is also necessary to point out that *mdx* mice in fact display a milder phenotype compared to DMD patients[66]. Muscle inflammation starts about 3 weeks of age, persists into 2–3 months of age, and then gradually subsides in limb muscles but not diaphragm. Our *Yy1/mdx* dKO mice however display more severe dystrophic pathologies mimicking human DMD. Therefore, CCL5/CCR5 axis constitutes a potential target for DMD treatment. Although gene and molecular

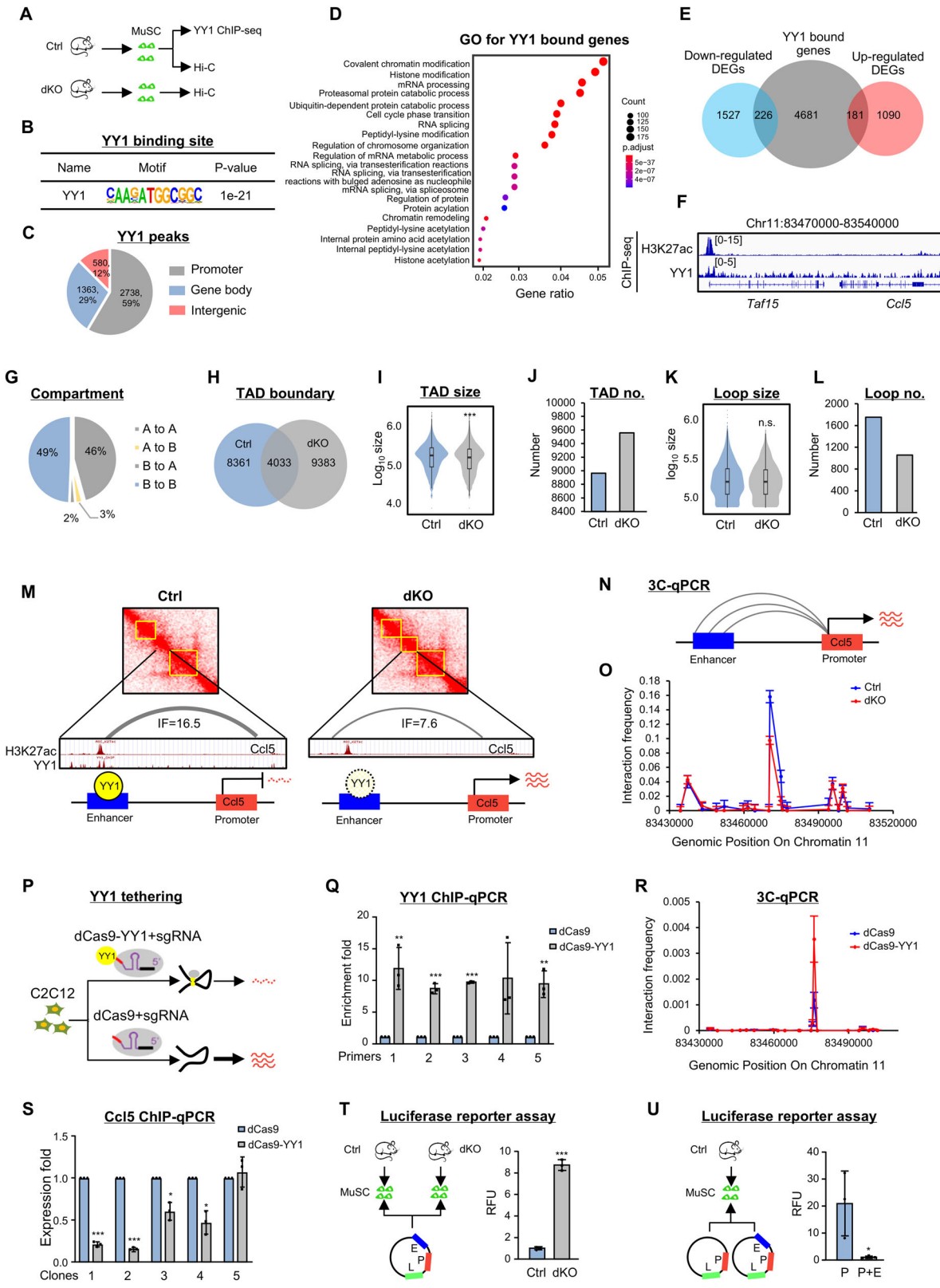

therapies targeting the primary defect of *Dystrophin* gene remain the most promising approach for DMD treatment, therapeutic strategies targeting the complex secondary mechanisms responsible for DMD pathogenesis are also being developed in parallel[22]. Drugs aiming at reducing inflammation and fibrosis are proven to be effective in mitigating the disease progression. In our study, MVC treatment leads to notable amelioration of the dystrophy pathologies; both

inflammation and fibrosis were reduced while muscle performance was improved, which encourages us to perform trials in human DMD patients in the future. In addition, our findings also reinforce the previously known pro-fibrotic role of TGFβ1 in dystrophic muscle highlighting it as a potential therapeutic target. Indeed, targeting TGFβ signaling by intramuscular injection of an inhibitor leads to reduced FAP accumulation and fibrosis in dystrophic mice[53].

**Fig. 6 | YY1 controls *Ccl5* expression in MuSC via regulating 3D looping interaction. A** Schematic of the experimental design for dissecting YY1 regulation of *Ccl5* expression. **B** Enrichment of YY1 motif in YY1 ChIP-seq binding regions. **C** Genomic distribution of 4681 YY1 binding peaks. **D** GO analysis of YY1 ChIP-seq target genes. **E** The overlapping between the above identified YY1 ChIP-Seq targets and the down/up-regulated genes. **F** Genomic snapshots on *Ccl5* locus showing co-binding of H3K27ac and YY1. **G** A/B compartment switch between Ctrl and dKO. **H–J** Comparison of TAD boundaries, size and number between Ctrl (*n* = 8965 TADs) and dKO (*n* = 9557 TADs). **K, L** Comparison of loop size and number between Ctrl (*n* = 1753 loops) and dKO (*n* = 1056 loops). The boxes (**I, K**) indicate median (center), Q25, and Q75 (bounds of box), the smallest value within 1.5 times interquartile range below Q25 and the largest value within 1.5 times interquartile range above Q75 (whiskers). **M** Heatmap and genomic snapshots showing the Hi-C interactions encompassing *Ccl5* locus (yellow box indicates TAD). **N, O** 3C-qPCR detection of E-P interaction on *Ccl5* locus, *n* = 3 cells. **P** Schematic of YY1 tethering experiment design in C2C12 cells. **Q** ChIP-qPCR detection of YY1 enrichment on the tethered site in dCas9-YY1 vs. dCas9 cells, *n* = 3 cells. *p* = 0.029, 0.0034, 0.000086, 0.10, 0.019. **R** 3C-qPCR detection of E-P interaction on *Ccl5* locus in the above cells, *n* = 3 cells. **S** RT-qPCR detection of *Ccl5* expression in the above cells, *n* = 3 cells. *p* = 0.00049, 0.000085, 0.023, 0.022, 0.57. **T** Left: Luciferase reporter assay design. Right: Relative fluorescence unit (RFU) of reporter activity is shown, *n* = 3 cells. *p* = 0.000015. **U** Left: Luciferase reporters assay design. Right: RFU of reporter activity is shown, *n* = 3 cells. *p* = 0.047. All the bar graphs are presented as mean ± SD, unpaired one-sided Student's *t* test (**I, K**), paired two-sided Student's *t* test (**Q, S**), unpaired two-sided Student's *t* test (**T, U**), one-sided Fisher's exact test (**D**) were used to calculate the statistical significance and adjustments were made for multiple comparisons: \**p* < 0.05, \*\**p* < 0.01, \*\*\**p* < 0.001, n.s. no significance. Source data are provided as a Source Data file.

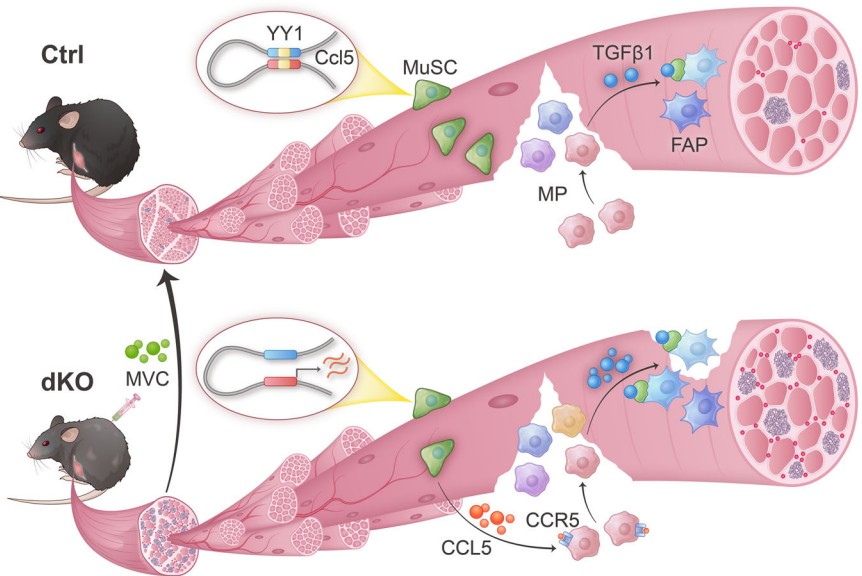

**Fig. 7 | Schematic illustration of how skeletal muscle stem cells modulate niche function in Duchenne muscular dystrophy.** In Ctrl MuSCs, YY1 binding to a *Ccl5* regulatory element orchestrates the E-P formation to suppress *Ccl5* expression. Upon YY1 loss in dKO MuSCs, Ccl5 is highly activated and augmented secretion of CCL5 from MuSCs promotes the recruitment of MPs via CCL5/CCR5 mediated cell crosstalk, which subsequently causes increased accumulation of TGFβ1 to hinder the apoptosis and clearance of FAPs therefore the aggravated fibrosis. MVC-mediated pharmacological blockade of the CCL5/CCR5 axis effectively mitigates muscle dystrophy and improves muscle performance in dKO mice.

Lastly, in search of reasons underlying *Ccl5* induction upon YY1 deletion, we demonstrate that YY1 acts on *Ccl5* enhancer to orchestrate E-P interactions thus reinforcing its role as a structural protein. Emerging evidence demonstrates YY1 contributes to E-P structural interactions in a manner analogous to DNA interactions mediated by CTCF[58]. It is shown that YY1 binds to active enhancers and promoter-proximal elements and forms dimers that facilitate the interaction of these DNA elements. Deletion of YY1 binding enhancer sites or depletion of YY1 protein disrupts E-P looping and gene expression[58]. These findings demonstrate the importance of enhancers in cell state transition and cell abundance[67]. In line with this, reduced E-P interaction frequency on *Ccl5* locus was observed in dKO MuSCs which was reversed by artificially tethering YY1 protein to the *Ccl5* enhancer region. Interestingly, our results suggest that the YY1-facilitated E-P interaction suppresses but not enhances *Ccl5* induction. Studies from others and ours in fact also demonstrate that binding of certain TFs renders an enhancer element to become suppressive in gene expression[58,68,69]. Additionally, we have only focused our investigation on *Ccl5*, it will also be interesting to elucidate how YY1 loss induces the expression of other immune genes and transforms the intrinsic epigenetic signaling to extrinsic regulation of the MuSC niche.

## Methods

### Mice

All animal handling procedures, protocols, and experiments ethics approval was granted by the CUHK AEEC (Animal Experimentation Ethics Committee) under Ref No. 21-080-GRF and 19-220-MIS. The mice were maintained in an animal room with 12 h light/12 h dark cycles, temperature (22–24 °C), and humidity (40–60%) at the animal facility in CUHK, fed with PicoLab® Select Mouse Diet 50 IF/9F Diet and provided with plenty of fresh clean water at all times. For all animal-based experiments, at least three pairs of littermates or age-matched mice were used.

*Yy1^f/f^* and C57BL/10 ScSn DMD*mdx* (*mdx*) mouse strains were purchased from The Jackson Laboratory (Bar Harbor, ME, USA). The YY1-inducible conditional KO (YY1^iKO^) strain (Ctrl: *Pax7^CreERT2/R26YFP^; Yy1^+/+^, Yy1^iKO^: Pax7^CreERT2/R26YFP^; Yy1^f/f^* mice) was generated by crossing Pax7^CreERT2/R26YFP^ mice with *Yy1^f/f^* mice. The *Yy1/mdx* double KO (YY1^dKO^) strain (Ctrl: *Pax7^CreERT2/R26YFP^; Yy1^+/+^; mdx*, YY1^dKO^: *Pax7^CreERT2/R26YFP^; Yy1^f/f^; mdx*) was generated by crossing YY1^iKO^ with *mdx* mice. Primers used for genotyping are shown in Supplementary Data 1.

### Animal procedures

Inducible deletion of YY1 was administered by IP injection of TMX (Sigma-Aldrich, T5648) at 100 mg/kg (body weight). For EdU (Sigma-

Aldrich, 900584-50MG) incorporation assay in vivo, 5 mg EdU (diluted in 100 µl PBS) injection via IP per day was performed for 3 consecutive days, followed by FACS isolation of MuSCs or FAPs 12 h later. MVC (Sigma-Aldrich, PZ0002-25MG) treatment was administrated by IP injection at 2 mg/kg every 2 days for 30 days. For voluntary wheel-running test, mice were housed individually for 7 days in polycarbonate cages with 12-cm-diameter wheels equipped with optical rotation sensors (Yuyan Instrument, ARW). For treadmill test, mice were adapted to a treadmill (Panlab, Harvard Apparatus, 76-0895) with a 5° incline at an initial speed of 10 cm/s, followed by a stepwise increase of 5 cm/s every two min until their exhaustion. For each animal experiment, Ctrl and YY1[dKO] mice of the same sex and age were used.

### Cell lines and cell culture
Mouse C2C12 myoblast cells (CRL-1772) and 293T cells (CRL-3216) were obtained from American Type Culture Collection and cultured in DMEM medium (Gibco, 12800-017) with 10% fetal bovine serum (Invitrogen, 16000044), 100 units/ml of penicillin and 100 µg of streptomycin (P/S, Gibco,15140-122) at 37 °C in 5% $CO_2$. All cell lines were tested negative for mycoplasma contamination.

### Fluorescence-activated cell sorting and culturing
MuSCs, fibro-adipogenic progenitors and macrophages were sorted based on established method[56,70–75]. Briefly, entire hindlimb muscles from mice were digested with collagenase II (Worthington, LS004177, 1000 units per 1 ml) for 90 min at 37 °C, the digested muscles were then washed in washing medium (Ham's F-10 medium (Sigma-Aldrich, N6635) containing 10% heat-inactivated horse serum (HIHS, Gibco, 26050088, 1% P/S)) before MuSCs were liberated by treating with Collagenase II (100 units per 1 ml) and Dispase (Gibco17105-041, 11 unit per 1 ml) for 30 min. The suspensions were passed through a 20 G needle to release cells. Mononuclear cells were filtered with a 40 µm cell strainer and sorted by BD FACSAria Fusion with the selection of the GFP+ (MuSCs); FITC-(CD45-, CD31-, ITGA7-) APC + (SCA1+) (FAPs); FITC-(Cd45-) APC-(Ly6G-) eFluor450 + (CD11b+) (MPs). Flowjo V10.8.1 was used for analysis of flow cytometry data. MuSCs were cultured in Ham's F10 medium with 20% FBS, 5 ng/ml β-FGF (Thermo Fisher Scientific, PHG0026) and 1% P/S, on coverslips and culture wells which were coated with poly-D-lysine solution (Sigma-Aldrich, p0899) at 37 °C overnight and then coated with extracellular matrix (ECM) (Sigma-Aldrich, E-1270) at 4 °C for at least 6 h. FAPs and MPs were cultured in DMEM medium with 10% FBS and 1% P/S.

### Isolation of bone marrow derived macrophages (BMDMs)
Isolating BMDM was performed according to literature[40,76] with slight modification. Briefly, bone marrow from adult *mdx* mouse was obtained by flushing femur and tibiae with DMEM and cells were cultured in DMEM containing 20% FBS and 20 ng/mL M-CSF (Thermo Fisher Scientific, PMC2044) for 6–7 days. The culture DMEM medium was changed on the 3rd day. After 7 days of culturing, cells were carefully washed by PBS twice and collected for experiments.

### Cell proliferation and apoptosis analyses
For EdU incorporation assay, freshly sorted MuSCs or FAPs were seeded into prepared coverslips and harvest immediately after adherence (in vivo) or cultured for days and added EdU to the culture medium for 6 h before harvest (in vitro). EdU staining was performed following the instruction of Click-iT® Plus EdU Alexa Fluor® 594 Imaging Kit (Thermo Fisher Scientific, C10639). Growing cells on coverslips were incubated with 10 µM EdU for a designated time before the fixation with 4% PFA for 20 min. EdU-labeled cells were visualized using "click" chemistry with an Alexa Fluor® 594-conjugated azide and cell nuclei were stained by DAPI (Life Technologies, P36931). Fluorescent images were captured with a fluorescence microscope (Leica). TUNEL (Terminal

deoxynucleotidyl transferase dUTP nick end labeling) assay was performed following our previous publication[12,77] and the instruction of In Situ Cell Death Detection Kit TMR red (Roche, 12156792910). Cells were cultured on coverslips for 36 h, followed by washing twice with PBS and fixing with 4% PFA for 15 min. TUNEL staining was carried out by adding reaction mixture of label solution for 30 min at dark. The coverslips were mounted with DAPI solution to stain the cell nucleus. Fluorescent images were captured with a fluorescence microscope (Leica).

### Co-culture assay
BMDMs or FAPs were re-suspended at the appropriate concentration in DMEM culture medium and seeded in the 24-well inserts (Corning, 353097, 353095). The bottom chamber was coated and seeded with MuSCs or MPs 1 day before the seeding of BMDMs or FAPs. After seeding and assembling, the co-cultured cells were incubated at 37 °C 5% $CO_2$ for 12–24 h and then inserts were harvested for experiments.

### Plasmids
For constructing the dCas9-YY1 plasmid, pAW91.dCas9 (Addgene, 104372) and pAW90.dCas9-YY1 (Addgene, 104373) plasmids were purchased from Addgene. Five sgRNAs were designed by CRISPOR[78] to target the *Ccl5* upstream enhancer region and cloned into a lentiguide-puro vector at the BsmbI site(lentiguide-puro-sgRNA1,2,3,4,5). For constructing the *Ccl5*-enhancer/promoter luciferase report plasmid, a 2456 bp DNA fragment of *Ccl5* enhancer region and 1500 bp *Ccl5* promoter were cloned into a pGL3-basic (purchased from Promega) vector between MluI and HindIII sites.

### Luciferase report assay
MuSCs were co-transfected with the *Ccl5*-enhancer/promoter luciferase reporter plasmid and internal control Renilla reporter plasmid. Cells were harvested 48 h after transfection through adding 100 µl lysis buffer and gently shaking for 15 min at room temperature. Luciferase activity was measured by the Dual-Luciferase kit (Promega, E1910) according to our previous publication[56,77]. The luminescent signal was recorded by Elmer VICTOR ™ X multilabel reader. The ratio of the reporter signal and the Renilla control signal was compared between different samples for further analysis.

### dCas9-YY1 tethering
For virus production, HEK293T cells were grown to 50–75% confluency on a 15 cm dish and then transfected with 15 µg of pAW91.dCas9 or pAW90.dCas9-YY1, 11.25 µg psPAX2 (Addgene, 12260), and 3.75 µg pMD2.G (Addgene, 12259). Viral supernatant was cleared of cells by filtering with 0.2 µm filter membrane (Pall Corporation, 4612). 5 mL of vital supernatant was mixed with 5 mL DMEM medium and added to C2C12 cells in the presence of polybrene (Santa Cruz Biotechnology, sc-134220) at 8 µg/mL. After 24 h, viral media was removed and fresh media containing blasticidin (Thermo Fisher Scientific, R21001) at 10 µg/mL. Cells were selected until all cells on non-transduced plates died, to obtain the blasticidin+ dCas9 C2C12 and dCas9-YY1 C2C12 cell lines. The tethering guide RNAs were packaged by the virus as described above using the lentiguide-puro-sgRNA1,2,3,4,5 plasmids and were transduced into dCas9 C2C12 and dCas9-YY1 C2C12 cells respectively. After 24 h, viral media was removed and fresh media containing puromycin (Thermo Fisher Scientific, A1113802) at 2.5 mg/mL. Cells were selected until all cells on non-transduced plates died. Double-positive cells (blasticidin+, puromycin +) were identified and expanded.

### RNA extraction and real-time PCR
Total RNAs were extracted using TRIzol reagent (Invitrogen, 15596026) following the manufacturer's protocol. For quantitative RT-PCR, cDNAs were reverse transcribed using HiScript III First-Strand

cDNA Synthesis Kit (Vazyme, R312-01). Real-time PCR reactions were performed on a LightCycler 480 Instrument II (Roche Life Science) using Luna Universal qPCR Master Mix (NEB, M3003L). Sequences of all primers used can be found in Supplementary Data 1.

## Immunoblotting, immunofluorescence, and immunohistochemistry

For Western blot assays, according to our prior publication[79–81], cultured cells were washed with ice-cold PBS and lysed in cell lysis buffer. Whole cell lysates were subjected to SDS–PAGE and protein expression was visualized using an enhanced chemiluminescence detection system (GE Healthcare, Little Chalfont, UK) as described before[74]. The following dilutions were used for each antibody: YY1 (Abcam ab109237; 1:1000), CCL5 (Abcam ab189841; 1:500), CCR5 (Abcam ab65850; 1:500), TGF-β1 (Abcam ab92486; 1:1000), α-TUBULIN (Santa Cruz Biotechnology sc-23948; 1:5000), GAPDH (Sigma-Aldrich G9545-100UL; 1:5000), CAS9 (CST 14697 T; 1:1000). For immunofluorescence staining, cultured cells were fixed in 4% PFA for 15 min and blocked with 3% BSA within 1 h. Primary antibodies were applied to samples with indicated dilution below and the samples were kept at 4 °C overnight. For immunofluorescence staining[12,75], cultured cells were fixed in 4% PFA for 15 min and permeabilized with 0.5% NP-40 for 10 mins. Then cells were blocked in 3% BSA for 1 h followed by incubating with primary antibodies overnight at 4 °C and secondary antibodies for 1 h at RT. Finally, the cells were mounted with DAPI to stain the cell nucleus and images were captured by a Leica fluorescence microscope. Primary antibodies and dilutions were used as following PAX7 (Developmental Studies Hybridoma Bank; 1:50), YY1 (Abcam ab109237, 1:200), CCL5 (Abcam ab189841; 1:200), F4/80 (Abcam ab6640, 1:200), PDGFRα (R&D BAF1062; 1:200), Ki67 (Santa Cruz Biotechnology, sc-23900; 1:200). For immunohistochemistry[12,72,75], in brief, slides were fixed with 4% PFA for 15 min at room temperature and permeabilized in ice-cold methanol for 6 min at −20 °C.Heat-mediated antigen retrieval with a 0.01 M citric acid (pH 6.0) was performed for 5 min in a microwave. After 4% BSA (4% IgG-free BSA in PBS; Jackson, 001-000-162) blocking, the sections were further blocked with unconjugated AffiniPure Fab Fragment (1:100 in PBS; Jackson, 115-007-003) for 30 min. The biotin-conjugated anti-mouse IgG (1:500 in 4% BBBSA, Jackson, 115-065-205) and Cy3-Streptavidin (1:1250 in 4% BBBSA, Jackson, 016-160-084) were used as secondary antibodies. Primary antibodies and dilutions were used as follows: CCL5 (Abcam ab189841; 1:200), CCR5 (Abcam ab65850; 1:200), TGF-β1 (Abcam ab92486; 1:200), PDGFRα (R&D BAF1062; 1:200), Collagen Ia1(Novus NBP1-30054; 1:200), F4/80 (Abcam ab6640), CD68 (Biorad MCA1957GA; 1:200), CD206 (Abcam ab64693;1:200), Laminin (Sigma-Aldrich L9393-100UL, 1:800), α-SMA (Invitrogen, 14-9760-82; 1:200), Ki67 (Santa Cruz Biotechnology sc-23900; 1:200); PAX7 (Developmental Studies Hybridoma Bank; 1:50), eMyHC (Leica NCL-MHC-d; 1:200) for staining of muscle cryosections. Images were slightly modified with ImageJ in which background was reduced using background subtraction and brightness and contrast were adjusted. H&E (Hematoxylin and eosin), was performed as previously described[12,74,82]. Masson's trichrome staining was performed according to the manufacturer's (ScyTek Laboratories, Logan, UT) instructions. Oil Red O staining was performed according to the manufacturer's (Abcam, ab150678) instructions.

## Single-cell RNA-seq and data analysis

Single-cell RNA-seq was performed on 10x genomics platform. Briefly, whole muscle cells were isolated with an additional step of viability validation by Propidium Iodide (PI) staining. Red blood cells were eliminated by ACK buffers (150 M $NH_4Cl$, 100 mM $KHCO_3$, 10 mM EDTA-2Na) before sorting. After sorting, live cells were washed with 0.04% BSA in PBS twice and resuspended in the BSA solution at an appropriate concentration (300–1200 cells/μl). Suspended cells were

counted under a microscope and Typan blue was used to examine the cell viability. After the confirmation of cell number and viability, library construction was performed following the manufacturer's instructions for generation of Gel Bead-In Emulsions (GEMs) using the 10x Chromium system. The single cell RNA library was sequenced by Illumina HiSeq X Ten instrument. CellRanger 7.0.0 and Seurat 4.3.0 were used to analyze the single-cell data. Ctrl and dKO groups were initially merged together and filtered with quality control parameters (cells with more than 10% expression on mitochondrial genes, or fewer than 500 total features expressed were filtered out and 7.5% doublets were removed according to captured cell numbers; Supplementary Fig. 2D–I). The two groups were integrated using FindIntegrationAnchors and IntegrateData function wrapped in Seurat package to minimize batch effects[32,43,46]. Dimensionality reduction was performed through Principal Component Analysis. UMAP embedding parameters were based on the top 40 PCs and embedded in 2-dimensions to visualize the data. Cells from Ctrl and dKO groups were relatively evenly distributed in all clusters, indicating no major batch effect after treatment (Supplementary Fig. 2J). To annotate different cell types, differentially expressed genes among cell clusters were identified using FindAllMarkers function. Genes were identified as significantly differentially expressed if FDR < 0.05 and expression in at least 20% of cells. To dissect the subtypes of FAPs and macrophages, we conducted second-round clustering (sub-clustering) within each cell type. Differentially expressed gene markers were also examined for subclusters for manual subtype annotation. Monocle 2.22.0 was used for pseudotime trajectory analysis[83]. The raw counts data from FAP and macrophage populations were used for trajectory inference and the top sub-cluster DEGs were used as input gene lists for trajectory construction analysis. AddModuleScore function wrapped in Seurat was used to calculate the average expression levels of each gene set of interest (Supplementary Data 2) at single-cell level, yielding module scores named as anti-apoptotic, apoptotic score and inflammatory score.

## Bulk RNA-seq and data analysis

For RNA-seq (polyA + mRNA)[72,73], total RNAs were subjected to polyA selection (Ambion, 61006) followed by library preparation using NEBNext Ultra II RNA Library Preparation Kit (NEB, E7770S). Libraries were paired-end sequenced with read lengths of 150 bp on Illumina HiSeq X Ten or Nova-seq instruments. The raw reads of RNA-seq were processed following the procedures described in our previous publication[56]. Briefly, the adapter and low-quality sequences were trimmed from 3′ to 5′ ends for each read, and the reads shorter than 36 bp were discarded. The clean reads were aligned to mouse (mm10) reference genome with STAR. Next, Cufflinks was used to quantify the gene expression. Genes with an expression level change greater than 1.5-fold and a $p$ value of <0.01 were identified as DEGs between two stages/conditions. GO enrichment analysis was performed using R package clusterProfiler.

## ChIP-seq and data analysis

YY1 ChIP was performed following our previously described protocol[12]. 10 μg of antibodies against YY1 (Santa Cruz Biotechnology, sc-1703), or normal mouse IgG (Santa Cruz Biotechnology, sc-2025) were used for immunoprecipitation. Immunoprecipitated genomic DNA was resuspended in 20 μl of water. For ChIP-seq DNA library construction, a NEBNext® Ultra™ II DNA Library Prep Kit for Illumina® (NEB, E7645S) was used according to the manufacturer's instructions. Bioanalyzer analysis and qPCR were used to measure the quality of DNA libraries including the DNA size and purity. Finally, DNA libraries were sequenced on the Illumina Genome Analyzer II platform. The raw data were first pre-processed by initial quality assessment, adapter trimming, and low-quality filtering and then mapped to the mouse reference genome (mm10) using bowtie2[84], and only the non-redundant

reads were kept. The protein DNA-binding peaks (sites) were identified using MACS293 with an input (IgG) sample as the background. During the peak calling, candidate peaks were compared with the background, dynamic programming was used to determine λ of Poisson distribution, and the P value cutoff was set to 0.0001 for YY1 ChIP-Seq experiment.

## Hi-C and data analysis

Hi-C was performed according to previously described protocols[56]. Libraries were prepared by on-bead reactions using the NEB Next Ultra II DNA Library Preparation Kit (NEB, E7645S). The beads were separated on a magnetic stand, and the supernatant was discarded. After washes, the beads were resuspended in 20 μl of 10 mM tris buffer and boiled at 98 °C for 10 min. The elute was amplified for 10 to 13 cycles of PCR with Phanta Master Mix (Vazyme, P511-01), and the PCR products were purified using VAHTS DNA Clean Beads (Vazyme, N411-01). The Hi-C libraries were paired-end sequenced with read lengths of 150 bp on an Illumina HiSeq X Ten instrument. Data were analyzed by Hi-C Pro, juicer box software and mapping to mouse genome mm10. Raw Hi-C data were processed as previously described[71]. Briefly, the in-situ Hi-C data was processed with a standard pipeline HiC-Pro[85]. First, adapter sequences and poor-quality reads were removed using Trimmomatic (ILLUMINACLIP: TruSeq3-PE-2.fa:2:30:10; SLIDINGWINDOW: 4:15; MINLEN:50). The filtered reads were then aligned to mouse reference genome (mm10) in two steps: (1) global alignment was first conducted for all pair-end reads, (2) the unaligned reads were split into prospective fragments using restriction enzyme recognition sequence (GATCGATC) and aligned again. All aligned reads were then merged and assigned to restriction fragments, while low quality (MAPQ < 30) or multiple alignment reads were discarded. Invalid fragments including unpaired fragments (singleton), juxtaposed fragments (re-ligation pairs), unligated fragments (dangling end), self-circularized fragments (self-cycle), and PCR duplicates were removed from each biological replicate. The remaining validated pairs from all replicates of each stage were then merged, followed by read depth normalization using HOMER and matrix balancing using iterative correction and eigenvector decomposition (ICE) normalization to obtain comparable interaction matrix between different stages.

Following previous procedure[86], to separate the genome into A and B compartments, the ICE normalized intra-chromosomal interaction matrices at 100-kb resolution were transformed to observe/expect contact matrices, and the background (expected) contact matrices were generated to eliminate bias caused by distance-dependent decay of interaction frequency and read depth difference[56,71]. Pearson correlation was then applied to the transformed matrices and the first principal component (PC1) of these matrices was divided into two clusters. The annotation of genes and the expression profile were used to assign positive PC1 value to gene-rich component as compartment A and negative PC1 value to gene-poor component as compartment B.

Normalized contact matrix at 10 kb resolution of each time point was used for TAD identification using TopDom[87]. In brief, for each 10-kb bin across the genome, a signal of the average interaction frequency of all pairs of genome regions within a distinct window centered on this bin was calculated, thus TAD boundary was identified with local minimal signal within certain window. The falsely detected TADs without local interaction aggregation were filtered out by statistical testing. Invariant TADs were defined using following criteria: (1) the distance of both TAD boundaries between two conditions is no more than 10 kb; (2) the overlapping between two TADs should be larger than 80%; stage-specific TADs were defined otherwise. Loops are identified by using HiCCUPS module of Arrowhead with default parameter at 10 kb resolutions.

## Quantitative analysis of chromosome conformation capture assays (3C-qPCR)

3C-qPCR was performed following published protocols[88]. The chromatin was cut by HindIII restriction enzyme to obtain DNA fragments. The promoter region of Ccl5 was set as the anchor to detect the designed E-P interactions. Primers were designed targeting the nearby site of Ccl5 enhancer region. 18s rRNA was used as an internal control for quantification normalization. Sequences of the oligos used in the study were included in Supplementary Data 1.

## Statistics and reproducibility

Data represent the average of at least three independent experiments or mice ± s.d. unless indicated. No statistical method was used to predetermine sample size. "No data were excluded from the analyses. The statistical significance of experimental data was calculated by the two-sided paired Student's t test (Fig. 1B, I, J; Fig. 3E, I, K; Fig. 4J, L, N; Fig. 6Q, S), two-sided unpaired Student's t test (Fig. 1E–H, K, L, N–R; Fig. 2B–D, K, N; Fig. 3F, G, L–N; Fig. 4B–F, H, I, O, Q; Fig. 5B–I, K; Fig. 6E–H, K, L, N–R), one-sided unpaired Student's t test one-sided (Figs. 3B; 6I, K), one-sided Fisher's exact test and adjustments were made for multiple comparisons (Fig. 3C, D; Fig. 6D; Supplementary Data 2–4): *p < 0.05, **p < 0.01, ***p < 0.001 and n.s.: no significance p ≥ 0.05. Specifically, a single zero-truncated negative binomial distribution was fitted to the input data and each region was assigned a P value based on the fitted distribution. Representative images of at least five independent experiments are shown in FigS. 1D, E, H, I–L O, P; 3F, J; 4B–F, H, I, L, Q; 5B, E, G, H; and Supplementary Fig. 1B, C, J–L, O; 3H; S4A–J; 5A, E.

## Reporting summary

Further information on research design is available in the Nature Portfolio Reporting Summary linked to this article.

## Data availability

In situ, Hi-C, ChIP-seq, bulk RNA-seq, and scRNA-seq data reported in this paper are deposited in the Gene Expression Omnibus database under accession GSE250204. All data supporting the findings of this study are provided with Source data. Source data are provided with this paper.

## Code availability

The code used in this study is available at the GitHub repository https://github.com/Hannah-bioinfo/Scripts_for_YY1_paper.

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

## Acknowledgements

We would like to thank Home for Researchers (www.home-for-researchers.com) for assistance in drawing. This work was supported by National Key R&D Program of China [2022YFA0806003 to H.W.]; National Natural Science Foundation of China [82172436 to H.W.]; General Research Fund (GRF) from the Research Grants Council (RGC) of the Hong Kong Special Administrative Region, China [14106521, 14100620, 14105823 and 14115319 to H.W.;14103522, 14105123 and 14120420 to H.S.]; Theme-based Research Scheme (TRS) from RGC [T13-602/21-N to H.W.]; Strategic Topics Grant (STG) from RGC [STG1/E-403/24-N to H.W.]; Area of Excellence Scheme (AoE) from RGC [AoE/M-402/20 to H.W.]; Health and Medical Research Fund (HMRF) from Health Bureau of the Hong Kong Special Administrative Region, China [10210906 and 08190626 to H.W.]; the research funds from Health@InnoHK program launched by Innovation Technology Commission, the Government of the Hong Kong SAR, China [to H.W.]; Chinese University of Hong Kong (CUHK) Strategic Seed Funding for Collaborative Research Scheme (SSFCRS) [to H.W.].

## Author contributions

Yang Li designed and performed most of experiments; Fengyuan Chen and Yeelo Cheung performed and helped with animal experiments; Chuhan Li and Qiang Sun analyzed RNA-seq, ChIP-seq, Hi-C, and scRNA-seq data; Yu Zhao supervised Hi-C assay; Xingyuan Liu helped the revision of the experiments and results; Bénédicte Chazaud provided constructive suggestions and supervised co-culture experiments; Ting Xie contributed to the manuscript writing; Hao Sun supervised computational analyses; Yang Li and Huating Wang wrote the manuscript, with inputs from all authors.

## Competing interests

The authors declare no competing interests.
