## [Peer Review File · Nature Communications]

REVIEWER COMMENTS

Reviewer #1 (Remarks to the Author):

In the manuscript titled "Skeletal muscle stem cells modulate niche function in Duchenne muscular dystrophy (DMD) through YY1-CCL5 axis," Li et al. demonstrate the role of the transcription factor YY1 in muscle stem cell behavior and its influence on the interaction between muscle stem cells and their surrounding environment. The study provides insights into how muscle stem cells (MuSCs) can impact their microenvironment, the role of YY1 in DMD progression, and the molecular mechanism through which MuSCs communicate with macrophages via CCL5. However, concerns arise regarding the novelty of the data, as the same authors have previously published similar findings. Additionally, the low number of animals used in various experiments raises questions about the robustness of the findings, particularly in the context of a complex disease phenotype. Furthermore, though the proposed MuSCs-Ccl5-Macrophages-Tgfb1-FAP mechanism shows promise, there are still many unanswered questions surrounding this mechanism. Despite the authors' commendable efforts to explore intercellular communication during muscle regeneration, the manuscript, in its current form, falls short of meeting the standards of novelty and impact expected for publication in Nat Com.

Major concerns:

1) Novelty of the identification of YY1 on MuSC behavior, especially for Figure 1. From this group, Chen et al., have already described and characterized the conditional YY1 deletion from MuSCs and the consequences of YY1 deletion on muscle regeneration after an acute and repetitive injuries, as well as in the mdx mouse. In fact, Figure 1 of this manuscript has the same title and shows very similar data as Figure 3 in their previous paper (<https://www.embopress.org/doi/full/10.15252/embj.201899727>). The authors did not mention this in their introduction, nor did they discuss this. Thus, novelty and, thereby, impact, two crucial criteria for Nat Com, cannot be established.

2) One major concern with the data presented is that some figures do not show individual numbers (N), and no standard deviation (or SEM). These figures are 1B, 1G, 1H; 3E, 3I, 3J; 4K, 4O, 4P; 6J, 6L, 6Q. And supplementary figures: S1F, S1H, S1L, S1M; S3A, S3C, S3D, S3E. Therefore, it is unclear whether there is only 1 animal in this cohort, which would hinder the interpretation of the data. Additionally, Figure 4O and P have only an n=2, with which formal statistical analysis cannot be performed.

3) Similarly, throughout the ms, the authors relied on very low Ns. Sometimes only n=3. For complex mouse alleles such as being used here in this paper, this can be challenging. Many results might be different if more animals were included (i.e. 1N, 1Q, 2 M, 2N, 3E, 3H, 3I, 3O, 3P, all of Figure 5). This is particularly concerning for Figure 5.

4) Concerns related to Figure 4:

a) Figures 1 & 2 suggest that there is an increase in FAPs in the dKO mice between 10 and 50%. While FAPs from dKO mice seem to be resistant to apoptosis, which could explain the increase in numbers as the authors point out, this is in stark contrast to the EdU data, where FAPs from dKO mice are refractory to proliferate. A cell might be immortal, but how can it increase in numbers if it cannot proliferate? The authors might want to explore the cellular kinetics of FAPs at different time points post injury.

b) A decrease in proliferation could also indicate a higher differentiation rate (into either myofibroblasts or adipocytes).

c) Figure 4O & P and its interpretations are unsatisfactory. Clear explanation of what constitutes an inflammatory vs restorative macrophage is needed, as well as validation of the proposed hypothesis "...suggesting that the enriched TGFb accumulation in dKO muscle niche may be a direct result of increased number of MPs". The question remains, where is Tgfb1 coming from?

d) The fact that Tgfb1 seems to be only increased at 2 months would suggest that the effects on FAPs is also gradual. The authors, however, already saw a difference in FAP numbers at 21 days. Wouldn't that suggest that the impact of MuSCs-specific deletion of YY1 is independent of Tgfb1? Related to this, when do the authors see the first changes in Ccl5 expression/secretion? Based on Fig 2D, the impact of YY1 deletion is immediate on the MPs, which would mean that Ccl5 expression is also immediately altered. The question of timing is a very crucial one that needs answering. For example, the authors could investigate FAP dynamics during the early degenerative/regenerative phase, which would further our understanding of how YY1-MuSC-Ccl5, via macrophages, affect FAPs.

e) It is interesting how the Pdgfra staining in Figure 4F is nuclear, while the rest is localized to the membrane. As Pdgfra is a membrane receptor, this requires further clarification.

5) Concerns related to Figure 5: the Maraviroc data are the most exciting aspect of this paper. However, there are multiple concerns:

a. The low Ns, especially for the functional data, do not allow for any statistical power.

- b. The authors need to clarify the dose (2 or 20 mg/kg?). They should also indicate the concentration of DMSO (the vehicle) used.
- c. the authors also need to clarify if they treated the animals twice a day (described as “bi-daily” in main text) or every other day (method).
- d. it is not clear if the authors looked at the TA or DP in Fig 5.
- e. For 5B, the CSA results for the DMSO ctrl group is twice as the ctrl mice in Fig 1E display. Similarly, the DMSO dKO group also displayed ~30% larger CSA compared to 1E. Is the DMSO treatment having any impact on myofiber stability?
- f. 5D&E, it appears as if the myofibers are larger and not smaller (based on Laminin staining) in the ctrl images for both the DMSO and MVC treated animals.

6) Wrong gene assignment used for scRNAseq: the authors list on page 10 the genes they used to assign FAP fates. Unfortunately, they used *Lpl* to assign fibrogenic and *Fn1* to adipogenic fates, which should have been the exact opposite. Therefore, the data in Fig 2 I & J are challenging to interpret.

7) The authors did not provide sufficient rationale for why they focused on YY1 from MuSCs. They only mention YY1 in 2 sentences in the introduction.

Minor concerns:

8) Primary literature citations are missing throughout the ms. Without references to support their statements and rationale, independent verification is challenging.

9) While exciting to explore and understand how MuSC and FAPs cross talk with each other, there is not sufficient data to support the hypothesis that MuSCs are directly impacting FAP behavior through YY1 as in vitro experiments show no impact of MuSC on FAPs (Figure 4G, H, I). In fact, the authors do show that the effect on FAPs is rather indirect through macrophages, which are altered when YY1 is removed. Thus, being specific is important.

10) When assessing overall muscle health, the authors focus on the average CSA of the fibers, noting that YY1-MuSC cKO animals have larger fibers. To further understand this phenotype, it would be helpful to see the distribution in fiber size, as well as total fibers (are they larger because only the larger ones survived?), as well as quantifying the amount and number of fibers with

centrally located nuclei, further characterizing the rate of degeneration/regeneration in the mdx background. These measurements would also be interesting to see following treatment with Maraviroc, and how it is affecting muscle during disease progression (is treatment preserving the dying smaller fibers, limiting the degeneration phase, increasing the regenerative phase?).

11) While treatment with Maraviroc is exciting and interesting, the authors state “Interestingly, even in Ctrl mice, the treatment led to slightly attenuated pathological fibrosis and inflammation”. However, there is no statistical difference between DMSO and Maraviroc treated Ctrl animals. Therefore, authors should refrain from overstating results. In fact, the authors use very strong wording (remarkable, significant, etc.) for sometimes small changes throughout the whole ms.

Reviewer #2 (Remarks to the Author):

Review of “Skeletal Muscle Stem Cells Modulate Niche Function in Duchenne Muscular Dystrophy through YY1-CCL5 Axis” by Yang Li et al.

This paper studies interactions between MuSC (Muscle Stem Cells), FAP (fibro-adipogenic progenitors), and MP (macrophage) cells in the stem cell niche and in response to muscle damage and repair. The authors utilized YY1 deletion in mdx (DMD (dystrophin) point mutation) mice (YY1 dKO) which have increased susceptibility to muscle damage. A co-culture system was used to show that MuSCs seem to interact more directly with MPs than with FAPs in inducing an immune response in dystrophic muscle, and that MuSCs can contribute to MP recruitment via CCL5 and CCR5. The studies are clearly described and motivated and the results will be highly valuable to the tissue repair community.

An interesting result is that YY1 dKO mice have fewer MuSCs, and the perturbation seems to alter the balance of cell number or composition of cell types without changing too dramatically the function of either MuSC, FAP, or MP cell types. Clearly this balance is controlled by complex regulatory and signaling mechanisms, where YY1 is only one player, and the full networks of interactions are yet to be completely elucidated. However, it would quite useful in the discussion to make a connection between this cell number observation and recent work quantitatively modeling the somewhat simpler ESC differentiation to endoderm system (PMID: 37488417), which knocked out individual enhancers by CRISPRi and came to the conclusion that “these enhancers may

contribute to cell state transitions and cell abundance while not strongly affecting transcript levels when tested in the established state. This suggests a direct mechanism by which enhancers may contribute to human disease even in the absence of strong effects in post-transition cells.” Although the ESC-DE study perturbed individual enhancers, it is likely that YY1 dKO is also perturbing a subset of enhancers.

Additional typos:

cross talks → crosstalk

hinted its → hinted at its

Reviewer #3 (Remarks to the Author):

This manuscript by Wang H and colleagues is an important follow-up study of their Chen et al (2019, EMBO J) paper on the cell-autonomous role of YY1 protein in regulating muscle stem cells (satellite cells) and muscle regeneration. Here, Li et al “Skeletal muscle stem cells modulate niche function in Duchenne Muscular Dystrophy through YY1-CCL5 axis” report a non-cell-autonomous role of YY1 in satellite cells that subsequently regulate macrophage infiltration and muscle inflammation in a mouse model of muscular dystrophy. They utilized series of sequencing techniques and disease modeling to substantiate their conclusions. In the manuscript, the authors proposed that Yy1 deletion in muscle stem cells in mdx mouse model leads to an augmented inflammatory muscle environment characterized by increased number of macrophages and FAPs. The macrophages are recruited by the “CCL5 (satellite cells) – CCR5/TGFβ (macrophages) –FAP apoptosis” axis. In this scenario, there is an elevated level of TGFβ in YY1/Dmd DKO muscle that promotes the survival of (fibrotic?) FAPs by suppressing the timely apoptosis of FAPs that are required for the efficient regeneration to proceed. The authors showed that the inflammatory environment can be alleviated using CCR5 inhibitor in the DKO mouse model. The authors further showed that YY1 inhibits the expression of CCL5 in MuSC after forming the enhancer-promoter looping at the Ccl5 locus. Loss of YY1 leads to reduced E-P looping at the Ccl5 locus which favors the expression of Ccl5.

The data largely support the conclusions. However, several issues should be clarified to solidify the conclusion. First, the significance of the induction of CCL5 protein expression in MuSCs is questionable, as MuSCs are reduced by 60% while MPs increased by 250% in DKO (and MPs are known to express CCL5). The relative source of CCL5 (satellite cells vs MPs) and time of expression (i.e. if Satellite cell express CCL5 before MP infiltration?) should be addressed. The CCR5 inhibitor injection blocks the overall CCL5-CCR5 signaling, which can be contributed both from MuSCs and MPs. Moreover, it is quite unexpected that the enhancer-promoter looping facilitated by YY1

actually leads to the inhibition of the transcription of Ccl5. In fact, the result interpretation in the experiments aimed to demonstrate the “enhancer” region might actually be the promoter region of Taf15 gene.

Major concerns:

1. In Figure 1M-1R, the exercise performance of the dKO mice were examined at the age of 8.5M, yet almost 50% of the dKO mice cannot survive past 6M. What is the rationale of examining the phenotype at such a late stage when mice are dying?
2. Figure 1S, what is the survival curve of dKO mice after ~160 days? Based on the curve, no more dKO were found reduced?
3. Figure 1M, from the picture, the dKO mice were less than half of the Ctrl size yet the body weight was similar (Figure 1O). Is this due to lack of statistical power?
4. Figure 2B: how do the authors explain the decrease of MP at 60 days after TMX injection in the dKO mice compared to the Ctrl?
5. Figure 2M, in the manuscript, the authors described a population of MHC II+ cells in the dKO sample while in the figure, the #3 population is named resident. Are these the same cells? What is the difference in gene expression in these two populations?
6. Figure 3E-3G: the authors did RT-qPCR and ELISA to check the relative abundance of Ccl5 mRNA and protein in the MuSCs and showed there are roughly 2.5X increases in the expression of Ccl5 in MuSCs. At the same time, they also showed that the TA muscle accumulate CCL5 at 2 mon (Figure 3H, appeared to be much more than 2.5X), and that MPs also express CCL5/CCR5. What is the timepoint of isolation of the MPs shown in Figure 3I? These results do not seem to be sufficient to conclude that it is the CCL5 secreted from MuSCs that attracts the MPs into the muscle. Especially the number of MuSCs decreased significantly already at 21 days post TMX injection.
7. Figure 3F, please show a co-stain/overlay image to show the identity of the MuSCs (the model that the authors used has a GFP reporter). What is the percentage of dKO MuSCs that express CCL5, related to Figure 3F.
8. Figure 3K, this is not strong evidence to support the interaction of MuSC and MPs since MPs also highly express CCL5.
9. Figure 4O-P, what is the timepoint of isolation of the MPs?
10. The authors proposed that the CCR5 expressing MPs are more enriched in the dKO due to the chemoattractant CCL5 expressed in the dKO. Is CCR5 expressing MPs accounting for the relative increase in the MPs in the dKO? What is the relative contribution of TGF β 1 expression from the CCR5 expressing MPs that are recruited to the dKO muscle?

11. The part where TGF β 1 secreted by the MPs which then influence the FAPs is not well-developed. Did the authors try to isolate MPs from Ctrl and dKO mice and then co-culture with the FAPs to check the effects on apoptosis, similar design as Figure 4G-H?
12. Figure 4: do CCL5 and TGF β 1 levels persist through in the dKO muscle? Did the authors check the level of CCL5 and TGF β 1 at different timepoints after TMX injection?
13. Figure 5: the authors used MVC, which is an inhibitor for CCR5 to try to reduce the recruitment of CCR5 expressing MPs into the dKO. Since this is IP injection, what is the effect of MVC on the overall MPs population? Does it decrease the CCR5 expressing MPs? This result shows that the CCR5 expressing MPs can aggravate the inflammatory phenotype in the mdx model, but not sufficient to conclude that CCL5 expressed by MuSC in the dKO are the driver for the recruitment of CCR5 expressing MPs.
14. Figure 5B: the timepoint of sampling for the MVC treatment was 2.5M after TMX injection. The CSA of the Ctrl mice receiving DMSO was around 2000 μ m². However, in Figure 1E, when the authors also quantified the CSA of the Ctrl muscle, the average CSA was around 1000 μ m². Why the average CSA of the Ctrl group (receiving vehicle control) so different in the two figures at the same timepoint?
15. Figure 5F-H: the authors showed that the relative abundances of the three populations were brought to comparable levels by CCR5 injection. What is the expression pattern of CCL5 in the dKO after MVC administration? Is TGF β 1 level reduced in the muscle?
16. Figure 6F: the authors proposed that YY1 does not directly bind to the promoter regions of CCL5, instead it can bind to an “enhancer” region. The genome track showed that the H3K27ac peaks seem to be at the promoter regions of the Taf15 gene? How does the authors reconcile this result?
17. Figure 6: If the YY1 is mediating the transcription of Taf15 gene, which is the so-called enhancer region according to the authors, then the interpretations are totally different. Figure 6O-R then means that the Taf15 promoter was active in the dKO compared to the Ctrl.
18. Figure 6T: as proposed in the manuscript, YY1 mediates the enhancer looping with promoter at the Ccl5 locus. Enhancers also possess low level transcription capability, did the authors take this into account? In addition, how does EP looping work in a circular DNA vector? Would it be possible to use a YY1-expressing plasmid in the luciferase assay which allows the authors to better pinpoint if YY1 can drive the transcription?
19. Figure 6U: what is the directionality of the Luciferase relative to the enhancer/promoter in the vector design? Depends on the design, this result can merely mean that the “enhancer” region can drive the transcription of the luciferase.
20. The overall figure layout is often redundant or un-necessary, and the placement of the individual figures doesn’t follow a logical order. For example, Figure 3H showed that CCL5 is increased in the muscle, yet immediately after that comes the MP expression pattern, and then CCL5/CCR5 expression pattern in TA muscle again. Followed by an IF staining. The order of the images does not

follow a good order. Another example is Figure 4M and 4N, why would the authors put the 2M data before the 7D data which showed no difference?

Minor concerns:

1. Title misleading and should specific information such as “mouse model of Duchenne muscular dystrophy”
2. There are many grammar mistakes throughout the manuscript.
3. Pp11, “while the stressed subset displayed a decrease (31% vs 23.2%)”, are the numbers in the wrong place since the comparison is dKO vs Ctrl?
4. The role of MuSCs secreting immune modulatory cytokines has been documented in other studies (eg. Zhu et al., PMID: 27524611). Claiming this finding as “unappreciated” appear to be exaggerating.
5. In Figure 2E, did the authors perform doublet-removal for the scRNA-seq analysis?
6. Figure 3L-3N, what is the rationale of using mdx mice derived BMDM for the migration assay?
7. Figure 4M, the staining of TGFb1 in the dKO group seems to be non-specific, especially the big bulge of the signal.
8. Figure 4F: what is the collection timepoint after EdU incorporation?
9. Figure 7 legends too brief. There is no information connecting CCR5 to TGFbeta.
10. Discussion on FAP subset affected as different FAPs have distinct functions – some promotes regeneration.

Reviewer #4 (Remarks to the Author):

Reviewer #5 (Remarks to the Author):

I co-reviewed this manuscript with one of the reviewers who provided the listed reports. This is part of the Nature Communications initiative to facilitate training in peer review and to provide appropriate recognition for Early Career Researchers who co-review manuscripts

Point to point response to reviewer comments:

Reviewer #1

In the manuscript titled "Skeletal muscle stem cells modulate niche function in Duchenne muscular dystrophy (DMD) through YY1-CCL5 axis," Li et al. demonstrate the role of the transcription factor YY1 in muscle stem cell behavior and its influence on the interaction between muscle stem cells and their surrounding environment. The study provides insights into how muscle stem cells (MuSCs) can impact their microenvironment, the role of YY1 in DMD progression, and the molecular mechanism through which MuSCs communicate with macrophages via CCL5. However, concerns arise regarding the novelty of the data, as the same authors have previously published similar findings. Additionally, the low number of animals used in various experiments raises questions about the robustness of the findings, particularly in the context of a complex disease phenotype. Furthermore, though the proposed MuSCs-Ccl5-Macrophages-Tgfb1-FAP mechanism shows promise, there are still many unanswered questions surrounding this mechanism. Despite the authors' commendable efforts to explore intercellular communication during muscle regeneration, the manuscript, in its current form, falls short of meeting the standards of novelty and impact expected for publication in Nat Com.

1.1 Novelty of the identification of YY1 on MuSC behavior, especially for Figure 1. From this group, Chen et al., have already described and characterized the conditional YY1 deletion from MuSCs and the consequences of YY1 deletion on muscle regeneration after an acute and repetitive injuries, as well as in the mdx mouse. In fact, Figure 1 of this manuscript has the same title and shows very similar data as Figure 3 in their previous paper (<https://www.embopress.org/doi/full/10.15252/emboj.201899727>). The authors did not mention this in their introduction, nor did they discuss this. Thus, novelty and, thereby, impact, two crucial criteria for Nat Com, cannot be established.

A: Thanks for the critical comments. It is true that the phenotypes of the dKO mice have been briefly characterized in our published paper(1) but the in-depth mechanistic dissection was not conducted. This was previously mentioned in the Results (page 8) and we have now also mentioned it in the Introduction (page 3).

In Fig. 3 of the published paper, we had only preliminarily dissected the phenotypes of YY1^{dKO} mice, but here in Fig. 1 we have performed a much more exhaustive characterization of the dKO phenotypes. We have confirmed the muscle regeneration defect (Fig.1E-G, Suppl. Fig. S1B-C), decreased MuSC pool and proliferation ability (Suppl. Fig. S1D-E), increased fibrosis (Fig. 1H, Suppl. Fig. S1F-G), a positive correlation of FAPs number and fibrosis (Fig. 1I-J), increased inflammation (Fig. 1K-L, Suppl. Fig. S1H-K), muscle loss (Fig. 1M-P), declined exercise capacity (Fig. 1Q-R) and lifespan (Fig. 1S).

More importantly, the scientific questions we ask and the findings from the current study are completely different from the published paper. In essence, the prior publication uncovers the intrinsic regulatory function/mechanism of YY1 in MuSCs while the current one depicts a MuSC launched extrinsic regulation network in chronic muscle regeneration. We therefore

argue that our study carries a high level of novelty and impact for publication in Nature Communications.

1.2 One major concern with the data presented is that some figures do not show individual numbers (N), and no standard deviation (or SEM). These figures are 1B, 1G, 1H; 3E, 3I, 3J; 4K, 4O, 4P; 6J, 6L, 6Q. And supplementary figures: S1F, S1H, S1L, S1M; S3A, S3C, S3D, S3E. Therefore, it is unclear whether there is only 1 animal in this cohort, which would hinder the interpretation of the data. Additionally, Figure 4O and P have only an n=2, with which formal statistical analysis cannot be performed. Similarly, throughout the ms, the authors relied on very low Ns. Sometimes only n=3. For complex mouse alleles such as being used here in this paper, this can be challenging. Many results might be different if more animals were included (i.e. 1N, 1Q, 2 M, 2N, 3E, 3H, 3I, 3O, 3P, all of Figure 5). This is particularly concerning for Figure 5.

A: Thanks for the critical comment. We apologize for the unclear presentation of our data. We have now included the information on numbers and SEM in all figures. We agree that n=3 may not be enough for some experiments, we have now repeated the experiments in main Fig. 1B, 1E-H, 1K, 1L, 1N-P; Fig. 3E, 3G, 3I, 3K-N; Fig. 4E, 4F, 4J, 4N; Fig. 5; and supplementary Fig. S1D, S1F, S1H, S1J-Q; Fig. S3A, S3C-E; Fig. S4; Fig. S5 with additional mice to increase the number to at least n=5. We have also shown standard deviation or SEM for all figures whenever needed.

1.3 Figures 1 & 2 suggest that there is an increase in FAPs in the dKO mice between 10 and 50%. While FAPs from dKO mice seem to be resistant to apoptosis, which could explain the increase in numbers as the authors point out, this is in stark contrast to the EdU data, where FAPs from dKO mice are refractory to proliferate. A cell might be immortal, but how can it increase in numbers if it cannot proliferate? The authors might want to explore the cellular kinetics of FAPs at different time points post injury. A decrease in proliferation could also indicate a higher differentiation rate (into either myofibroblasts or adipocytes).

A: Thanks for the critical comment. According to the suggestion, we have now closely examined the cellular kinetics of FAPs both in vitro and in vivo with more replicates and found that our original conclusion that dKO FAPs are refractory to proliferate may not be correct. Initially we have only performed EdU assays on FAPs obtained from in vitro cultured at one day and in vivo isolated at TMX-3D, with a sample size of n=3. We have now performed the assays both in vivo and in vitro at different time points after TMX injection, with n=5. For in vitro assay, FAPs were freshly isolated and cultured for 1, 2, or 3 days to detect the proliferation rate and no significant difference was observed between Ctrl and dKO groups at any time points (Fig. 4E, Suppl. Fig. S4A-B, page 14), which was consistent with our initial finding that the FAP proliferation rate was not affected in the dKO. In vivo EdU assays were performed at 5, 7 and 12 days after TMX injection. Similarly, no significant difference in EdU incorporation ratio was observed between Ctrl and dKO FAPs (Fig. 4F, Suppl. Fig.S4C-D, page 14). This is different from our initial finding showing dKO FAPs proliferated at a lower rate in vivo in dKO. We reason the new results are more reliable as more replicates and more time points were

used. Altogether, these findings have led us to conclude that deletion of YY1 in MuSCs does not affect the proliferation of FAPs, therefore, the anti-apoptotic ability could be the main reason why there is an increased number of dKO FAPs. Furthermore, as suggested, we have tested the potential of FAP differentiation into myofibroblasts or adipocytes. We found the fibrogenic differentiation of dKO FAPs was increased (Suppl. Fig. S4E and page 14) but the adipogenic difference was not affected (Suppl. Fig. S4F-G and page 14).

1.4 Figure 4O & P and its interpretations are unsatisfactory. Clear explanation of what constitutes an inflammatory vs restorative macrophage is needed, as well as validation of the proposed hypothesis “....suggesting that the enriched TGFb accumulation in dKO muscle niche may be a direct result of increased number of MPs”. The question remains, where is Tgfb1 coming from? The fact that Tgfb1 seems to be only increased at 2 months would suggest that the effects on FAPs is also gradual. The authors, however, already saw a difference in FAP numbers at 21 days. Wouldn't that suggest that the impact of MuSCs-specific deletion of YY1 is independent of Tgfb1? Related to this, when do the authors see the first changes in Ccl5 expression/secretion? Based on Fig 2D, the impact of YY1 deletion is immediate on the MPs, which would mean that Ccl5 expression is also immediately altered. The question of timing is a very crucial one that needs answering. For example, the authors could investigate FAP dynamics during the early degenerative/regenerative phase, which would further our understanding of how YY1-MuSC-Ccl5, via macrophages, affect FAPs.

A: Thanks for the critical comment. We acknowledge that in our initial submission, we did not do an exhaustive investigation on the TGFβ1 accumulation in dKO muscle niche. This is because prior publications have detailed how MPs secrete TGFβ1 to induce anti-apoptotic phenotype of FAPs resulting in their accumulation in chronically injured muscles(2) (page 6). Nevertheless, as suggested, we have now conducted additional experiments/analyses to further investigate the TGFβ1 role in dKO muscle niche. First, we have now combined the inflammatory and restorative MPs to show the TGFβ1 level from all MPs (original Fig. 4O-P and now Fig. 4O). This is because according to previous studies, most of the MPs in mdx muscles exhibited both inflammatory and restorative characteristics and there is distinct segregation among them(2). This was supported by analyzing our scRNA-seq data showing that MPs were the major contributor to TGFβ1 (Fig. 4M, page 15). We also found the level of TGFβ1 expression and secretion were similar in isolated MPs from dKO vs. Ctrl (Fig. 4N-O, page 15). Furthermore, Reviewer 3 has a similar question regarding the source of TGFβ1 and its effect on FAPs in his comment 3.11 below (page 11 of this file). By co-culture assay (Fig. 4Q, Suppl. Fig. S4J), we confirmed an increased number of MPs can lead to enhanced resistance of FAPs to apoptosis.

Also, to clarify the timing of TGFβ1 increase and its effect on FAPs number, we have now examined the expression of TGFβ1 at different time points (7, 14 days and 2 months after TMX injection). No significant difference was observed at TMX-7D (Suppl. Fig. S4H), which was consistent the data showing a similar FAP population at TMX-5D (Fig. 4C). However, a significant increase was detected later in dKO muscle at TMX-14D (Suppl. Fig. S4I) and TMX-2M (Fig. 4L), which was concomitant with elevated FAP number at TMX-21D and TMX-2M (Fig. 2C). Lastly, the high induction of CCL5 in dKO was observed immediately after TMX injection

(Fig. 3E-G), which was concomitant to the increased MPs at TMX-5D (Fig. 2D).

Altogether, the above findings support our conclusion that increased CCL5 level leads to a higher number of MPs which secrete more TGF β 1 to result in the gradual accumulation of FAPs in the dKO muscle.

1.5 It is interesting how the Pdgfra staining in Figure 4F is nuclear, while the rest is localized to the membrane. As Pdgfra is a membrane receptor, this requires further clarification.

A: Thanks for the critical comment. We believe this is caused by technical reasons. The staining in Figure 4F was conducted on freshly isolated FAPs immediately upon adherence to the culture surface, when the nuclear, cytoplasmic, and membrane portions could not be well separated yet. The staining in Figure 4E was performed on in vitro cultured FAPs where the cells were fully unfolded and PDGFR α was predominantly localized in the cytoplasm and membrane that can be separated from the nuclei.

1.6. Concerns related to Figure 5: the Maraviroc data are the most exciting aspect of this paper. However, there are multiple concerns. The low Ns, especially for the functional data, do not allow for any statistical power.

A: Thanks for the critical comment. We have now performed the experiment using additional pairs of mice to reach n=6 in Fig. 5B-E, 5G-J.

1.7 The authors need to clarify the dose (2 or 20 mg/kg?). They should also indicate the concentration of DMSO (the vehicle) used. the authors also need to clarify if they treated the animals twice a day (described as "bi-daily" in main text) or every other day (method).

A: We apologize for the incorrect description in the Results part. The dose for MVC was 2 mg/kg dissolved in 5% DMSO. The administration strategy was described in Fig. 5A that DMSO/MVC was injected every other day. We have now corrected the information on page 16.

1.8 it is not clear if the authors looked at the TA or DP in Fig 5.

A: Thanks for the critical comment. All the muscles in Fig. 5 are TA. We have now included the information on page 16.

1.9 For 5B, the CSA results for the DMSO ctrl group is twice as the ctrl mice in Fig 1E display. Similarly, the DMSO dKO group also displayed ~30% larger CSA compared to 1E. Is the DMSO treatment having any impact on myofiber stability?

A: Thanks for the critical comment. We cannot find any evidence of DMSO affecting muscle stability in the literature, we believe the observed difference may be caused by the sex difference as female mice were used in Fig. 1E and male mice in Fig. 5B. In addition, as required by your comments 1.2 and 1.15, we have now added more replicates and performed additional measurement (fiber size distribution, fiber number and percentage of central nuclei

fibers) to solidify the MVC treatment effect (Fig. 5B-D, page 16).

1.10 5D&E, it appears as if the myofibers are larger and not smaller (based on Laminin staining) in the ctrl images for both the DMSO and MVC treated animals.

A: Thanks for the critical comment. We apologize for using the non-representative images, they have now been replaced in Fig. 5D&E.

1.11 Wrong gene assignment used for scRNAseq: the authors list on page 10 the genes they used to assign FAP fates. Unfortunately, they used Lpl to assign fibrogenic and Fn1 to adipogenic fates, which should have been the exact opposite. Therefore, the data in Fig 2 I & J are challenging to interpret.

A: Thanks for the critical comment. We have now taken a closer look at the data. Our definition of FAP subsets was based on the most enriched genes and the trajectory fate, according to our re-analysis, we have now replaced *Lpl* to *Podn* to define fibrogenic FAPs (Suppl. Fig. S2L-M, page 10); similarly, we changed *Fn1* to *Igfbp5* to define adipogenic FAPs (Suppl. Fig. S2L-M, page 10).

1.12 The authors did not provide sufficient rationale for why they focused on YY1 from MuSCs. They only mention YY1 in 2 sentences in the introduction.

A: Thanks for the critical comment. As answered to your comment 1.1 (page 1 of this file), we have now revised the writing on page 3 of the Introduction to provide a better rationale for our study.

1.13 Primary literature citations are missing throughout the ms. Without references to support their statements and rationale, independent verification is challenging.

A: Thanks for the critical comment. We have now included all necessary primary citations throughout the manuscript. Please see the highlighted citations on pages 3-5, 21-23.

1.14 While exciting to explore and understand how MuSC and FAPs cross talk with each other, there is not sufficient data to support the hypothesis that MuSCs are directly impacting FAP behavior through YY1 as in vitro experiments show no impact of MuSC on FAPs (Figure 4G, H, I). In fact, the authors do show that the effect on FAPs is rather indirect through macrophages, which are altered when YY1 is removed. Thus, being specific is important.

A: Thanks for the critical comment. There may be some misunderstanding of our conclusion on MuSC and FAP crosstalk. Indeed, based on data in Fig 4G-I, we concluded that YY1 deletion in MuSCs has limited direct effects on FAPs. Please see the highlighted text on pages 14-15.

1.15 When assessing overall muscle health, the authors focus on the average CSA of the

fibers, noting that YY1-MuSC cKO animals have larger fibers. To further understand this phenotype, it would be helpful to see the distribution in fiber size, as well as total fibers (are they larger because only the larger ones survived?), as well as quantifying the amount and number of fibers with centrally located nuclei, further characterizing the rate of degeneration/regeneration in the mdx background. These measurements would also be interesting to see following treatment with Maraviroc, and how it is affecting muscle during disease progression (is treatment preserving the dying smaller fibers, limiting the degeneration phase, increasing the regenerative phase?).

A: Thanks for the critical comment. According to the suggestion, we have now measured the distribution in fiber size, total fibers as well as quantified the amount and number of fibers with centrally located nuclei, to characterize the rate of degeneration/regeneration in the *mdx* background in Fig. 1E-G (page 8). We found that YY1 specific deletion in MuSC leads to declined regeneration in *mdx* mouse muscle, with significantly decreased centrally located nuclei (CLN) fibers (Fig. 1F). Moreover, dKO mice showed enhanced muscle degeneration, with increased abnormally larger fibers (Fig. 1E) and decreased fiber numbers (Fig. 1G).

We also performed the above quantifications in Fig. 5B-D (page 16) to show the MVC treatment preserved the dying smaller fibers and inhibited the appearance of abnormally larger fibers (Fig. 5B), increased CLN fibers (Fig. 5C) and fiber numbers (Fig. 5D), indicating reduced degeneration and promoted regeneration.

1.16 While treatment with Maraviroc is exciting and interesting, the authors state “Interestingly, even in Ctrl mice, the treatment led to slightly attenuated pathological fibrosis and inflammation”. However, there is no statistical difference between DMSO and Maraviroc treated Ctrl animals. Therefore, authors should refrain from overstating results. In fact, the authors use very strong wording (remarkable, significant, etc.) for sometimes small changes throughout the whole ms.

A: Thanks for the critical comment. We now now revised the writing on page 16 and also toned down our statements throughout the text to avoid overstating.

Reviewer #2:

This paper studies interactions between MuSC (Muscle Stem Cells), FAP (fibro-adipogenic progenitors), and MP (macrophage) cells in the stem cell niche and in response to muscle damage and repair. The authors utilized YY1 deletion in mdx (DMD (dystrophin) point mutation) mice (YY1 dKO) which have increased susceptibility to muscle damage. A co-culture system was used to show that MuSCs seem to interact more directly with MPs than with FAPs in inducing an immune response in dystrophic muscle, and that MuSCs can contribute to MP recruitment via CCL5 and CCR5. The studies are clearly described and motivated and the results will be highly valuable to the tissue repair community.

2.1. An interesting result is that YY1 dKO mice have fewer MuSCs, and the perturbation seems to alter the balance of cell number or composition of cell types without changing too

dramatically the function of either MuSC, FAP, or MP cell types. Clearly this balance is controlled by complex regulatory and signaling mechanisms, where YY1 is only one player, and the full networks of interactions are yet to be completely elucidated. However, it would quite useful in the discussion to make a connection between this cell number observation and recent work quantitatively modeling the somewhat simpler ESC differentiation to endoderm system (PMID: 37488417), which knocked out individual enhancers by CRISPRi and came to the conclusion that “these enhancers may contribute to cell state transitions and cell abundance while not strongly affecting transcript levels when tested in the established state. This suggests a direct mechanism by which enhancers may contribute to human disease even in the absence of strong effects in post-transition cells.” Although the ESC-DE study perturbed individual enhancers, it is likely that YY1 dKO is also perturbing a subset of enhancers.

A: Thanks for the interesting comment. We have now cited the publication in the Discussion on page 23.

2.2. Additional typos: cross talks → crosstalk.

A: we have corrected the typos on page 6.

Reviewer #3

This manuscript by Wang H and colleagues is an important follow-up study of their Chen et al (2019, EMBO J) paper on the cell-autonomous role of YY1 protein in regulating muscle stem cells (satellite cells) and muscle regeneration. Here, Li et al “Skeletal muscle stem cells modulate niche function in Duchenne Muscular Dystrophy through YY1-CCL5 axis” report a non-cell-autonomous role of YY1 in satellite cells that subsequently regulate macrophage infiltration and muscle inflammation in a mouse model of muscular dystrophy. They utilized series of sequencing techniques and disease modeling to substantiate their conclusions. In the manuscript, the authors proposed that Yy1 deletion in muscle stem cells in mdx mouse model leads to an augmented inflammatory muscle environment characterized by increased number of macrophages and FAPs. The macrophages are recruited by the “CCL5 (satellite cells) – CCR5/TGFbeta (macrophages) –FAP apoptosis” axis. In this scenario, there is an elevated level of TGFβ in YY1/Dmd DKO muscle that promotes the survival of (fibrotic?) FAPs by suppressing the timely apoptosis of FAPs that are required for the efficient regeneration to proceed. The authors showed that the inflammatory environment can be alleviated using CCR5 inhibitor in the DKO mouse model. The authors further showed that YY1 inhibits the expression of CCL5 in MuSC after forming the enhancer-promoter looping at the Ccl5 locus. Loss of YY1 leads to reduced E-P looping at the Ccl5 locus which favors the expression of Ccl5.

The data largely support the conclusions. However, several issues should be clarified to solidify the conclusion. First, the significance of the induction of CCL5 protein expression in MuSCs is questionable, as MuSCs are reduced by 60% while MPs increased by 250% in DKO (and MPs are known to express CCL5). The relative source of CCL5 (satellite cells vs MPs) and time of

expression (i.e. if Satellite cell express CCL5 before MP infiltration?) should be addressed. The CCR5 inhibitor injection blocks the overall CCL5-CCR5 signaling, which can be contributed both from MuSCs and MPs. Moreover, it is quite unexpected that the enhancer-promoter looping facilitated by YY1 actually leads to the inhibition of the transcription of Ccl5. In fact, the result interpretation in the experiments aimed to demonstrate the “enhancer” region might actually be the promoter region of Taf15 gene.

3.1 In Figure 1M-1R, the exercise performance of the dKO mice were examined at the age of 8.5M, yet almost 50% of the dKO mice cannot survive past 6M. What is the rationale of examining the phenotype at such a late stage when mice are dying?

A: Thanks for the critical comment. There seems to be some misunderstanding of Fig. 1M-1R due to our unclear description in the original submission. The image shown in Fig. 1M was from the age of 8.5M but the experiments in Figure 1N-R were performed on the age 3.5M mice. We have now corrected the descriptions on page 9 to avoid confusion.

3.2 Figure 1S, what is the survival curve of dKO mice after ~160 days? Based on the curve, no more dKO were found reduced?

A: Thanks for the critical comment. The survival test duration was up to 6 months, at which time more than 50% of the dKO mice succumbed prior to this timeline. The remaining dKO mice exhibited notably subdued phenotypes, characterized by a significant reduction in body and muscle mass, alongside other ailments such as abnormal spine morphology and respiratory challenges (see below images). In adherence to the ethical regulation in our institute, we had to terminate the mice therefore were not able to continue the survival study after ~160 days.

3.3 Figure 1M, from the picture, the dKO mice were less than half of the Ctrl size yet the body weight was similar (Figure 1O). Is this due to lack of statistical power?

A: Thanks for the critical comment. As pointed out in the answer to comment 3.1 (page 8 of this file), Figure 1M shows the representative images of Ctrl and dKO mice at the age of 8.5-month-old. Due to the limited number of surviving dKO mice after 8.5M, we were not able to

collect enough number of mice for analysis. In Fig. 1N we used the 3.5-month mice for analysis and found a pronounced decrease in body weight in dKO vs. Ctrl, and this is statistically significant (n=5).

3.4 Figure 2B: how do the authors explain the decrease of MP at 60 days after TMX injection in the dKO mice compared to the Ctrl?

A: Thanks for the critical comment. We should point out that the decrease of dKO MP at 60 vs. 21 days after TMX injection in Fig. 2B was not statistically significant. Our main conclusion from this figure is the MP population was much higher in dKO vs. Ctrl muscles. We have now revised the writing on page 9 to clarify this point.

3.5 Figure 2M, in the manuscript, the authors described a population of MHC II+ cells in the dKO sample while in the figure, the #3 population is named resident. Are these the same cells? What is the difference in gene expression in these two populations?

A: Thanks for the critical comment. We apologize for the confusing annotation, the #3 population should be MHC II+ cells, we have made a correction in Fig. 2M.

3.6 Figure 3E-3G: the authors did RT-qPCR and ELISA to check the relative abundance of Ccl5 mRNA and protein in the MuSCs and showed there are roughly 2.5X increases in the expression of Ccl5 in MuSCs. At the same time, they also showed that the TA muscle accumulate CCL5 at 2 mon (Figure 3H, appeared to be much more than 2.5X), and that MPs also express CCL5/CCR5. What is the timepoint of isolation of the MPs shown in Figure 3I? These results do not seem to be sufficient to conclude that it is the CCL5 secreted from MuSCs that attracts the MPs into the muscle. Especially the number of MuSCs decreased significantly already at 21 days post TMX injection.

A: Thanks for the critical comment. We have now rearranged the panels in Fig. 3 to make it more logical for understanding. Fig 3H-J now depicts the expression of *Ccl5* and *Ccr5* in TA muscle while Fig. 3K (original 3I) depicts the *Ccr5* expression from MPs (*Ccl5* expression is removed from the figure due to its irrelevancy). We agree that the results from these figures only showed the correlation which is not sufficient to support MuSCs secreted CCL5 attracts MPs into the muscle. However, the subsequent co-culture assays provide more definitive evidence that MuSCs from dKO can secrete CCL5 which recruits MPs. Nevertheless, we cannot exclude the possibility that other cells such as T cells, MPs and muscle fibers in the dKO muscle may also secrete CCL5 into the muscle. For example, by analyzing scRNA-seq data we found that T cells can secrete CCL5 (data not shown). In terms of timing, The MPs in this figure were isolated 1-month post TMX injection. Despite the decreased number of MuSCs at the later stage (21-day post TMX injection), we think the recruitment process should have commenced promptly following TMX injection, evidenced by immediately elevated expression and secretion of CCL5 (Fig. 3E-G), along with escalated MPs in dKO muscles (Fig. 2D). We have now added the above discussion on page 21-22. We should also point out that Reviewer 1 has a related comment in 1.4 regarding the timing of the sequential regulatory events, please

see our answer to the comment above (page 3 of this file).

3.7 Figure 3F, please show a co-stain/overlay image to show the identity of the MuSCs (the model that the authors used has a GFP reporter). What is the percentage of dKO MuSCs that express CCL5, related to Figure 3F.

A: Thanks for the critical suggestions. We have now shown a co-staining image of the CCL5 and PAX7 in isolated MuSCs and quantified the CCL5+ MuSCs, which shows a significantly higher CCL5+ MuSCs in dKO (Fig. 3F and page 13).

3.8 Figure 3K, this is not strong evidence to support the interaction of MuSC and MPs since MPs also highly express CCL5.

A: Thanks for the critical comment. As stated in the answer to your comment 3.6 (page 9 of this file), data in Fig. 3K-N provides sufficient evidence to support the direct interaction of MuSCs and MPs. But we agree other cells such as T cells, MPs and muscle fibers can also express CCL5. We have now revised the writing on page 13 and 21 to further clarify this point.

3.9 Figure 4O-P, what is the timepoint of isolation of the MPs?

A: Thanks for the critical question. The original Fig. 4O-P depicted the expression of TGF β 1 in isolated inflammatory and restorative MPs (1 month after TMX injection), respectively. We have now replaced them with Fig. 4N which depicts the expression of *Tgf β 1* in total MPs. As stated in the answer to Reviewer 1's comment 1.4 (page 3 of this file), there is no clear segregation of inflammatory and restorative MPs in dystrophic muscles(2). By analysis our scRNA-seq data, over 70% of MPs expressed *Tgf β 1*.

3.10 The authors proposed that the CCR5 expressing MPs are more enriched in the dKO due to the chemoattractant CCL5 expressed in the dKO. Is CCR5 expressing MPs accounting for the relative increase in the MPs in the dKO? What is the relative contribution of TGF β 1 expression from the CCR5 expressing MPs that are recruited to the dKO muscle?

A: Thanks for the critical comment. There seems to be some misunderstanding of our results in Fig. 3. We did not suggest the appearance of CCR5+ MPs. In fact, we believe CCR5 expression increased in all MPs as a consequence of increased CCL5 expression in the dKO muscle niche; this is consistent with a previous publication showing enriched ligand microenvironment promotes the up-regulated receptor expression in cells(4). In line with this, co-staining of CCR5 and F4/80 on isolated MPs indeed showed all F4/80 positive MPs exhibited CCR5 expression in both Ctrl and dKO (Suppl. Fig. S3H, page 13). Analysis of the scRNA-seq and RT-qPCR (Suppl. Fig. S3F-G, page 13) also showed elevated *Ccr5* expression in all dKO MPs. As for the source of TGF β 1, we have elucidated in the answer to your comment 3.9 (page 10 of this file) and Reviewer 1's comment 1.4 (page 3 of this file) that the increased TGF β 1 expression should be contributed by all the MPs. We have now revised the writing on page 13 to clarify the point.

3.11 The part where TGFβ1 secreted by the MPs which then influence the FAPs is not well-developed. Did the authors try to isolate MPs from Ctrl and dKO mice and then co-culture with the FAPs to check the effects on apoptosis, similar design as Figure 4G-H?

A: Thanks for the critical comments and suggestions. Reviewer 1 has a similar comment in his/her comment 1.4 (page 3 of this file). We agree that the MP-FAP interaction via TGFβ1 was not well developed in our original submission. This is because prior publications have detailed how MPs secrete TGFβ1 to induce anti-apoptosis of FAPs resulting in their accumulation in chronically injured muscles(2). As suggested, we have now isolated MPs from Ctrl and dKO mice and co-cultured with FAPs (isolated from Ctrl mice). We found that when seeding the same number of Ctrl and dKO MPs no significant difference was observed in FAP apoptosis (Fig. 4P-Q, page 15). But when 5: 3 of dKO:Ctrl MPs were seeded mimicking the in vivo situation, a significant decrease of FAP apoptosis was observed in dKO group (Suppl. Fig. S4J, page 15). The above results indicate that the heightened anti-apoptotic ability identified in dKO FAPs is associated with the quantity of co-cultured MPs, rather than the source (Ctrl or dKO). In addition, according to Reviewer 1's suggestion, we have now examined the expression of TGFβ1 at different time points (7, 14 days and 2 months after TMX injection) to show that the timing of TGFβ1 increase is in correlation with FAP increase (Fig. 4L, Suppl. Fig. S4H-I). Altogether, our findings demonstrate that an increased number of MPs can lead to enhanced secretion of TGFβ1, thereby causing FAP resistance to apoptosis.

3.12 Figure 4: do CCL5 and TGFβ1 levels persist through in the dKO muscle? Did the authors check the level of CCL5 and TGFβ1 at different timepoints after TMX injection?

A: Thanks for the critical questions. Reviewer 1 has a similar comment in 1.4 regarding the timing of the events (page 3 of this file). In fact, we checked the CCL5 expression in 7D, 1M, and 2M after TMX injection in muscle in the original submission (Fig. 3H, page 13) and found a gradually increased accumulation of CCL5 levels in the dKO muscles. Additionally, TGFβ1 was also checked in 7D, 14D and 2M after TMX injection (Fig. 4L, Suppl. Fig. S4H-I, page 15); no significant changes were observed at 7D but substantial increases were observed in dKO muscles at TMX-14D and TMX-2M, indicative of gradual and persistent accumulation of TGFβ1 within the dKO muscle microenvironment.

3.13 Figure 5: the authors used MVC, which is an inhibitor for CCR5 to try to reduce the recruitment of CCR5 expressing MPs into the dKO. Since this is IP injection, what is the effect of MVC on the overall MPs population? Does it decrease the CCR5 expressing MPs? This result shows that the CCR5 expressing MPs can aggravate the inflammatory phenotype in the mdx model, but not sufficient to conclude that CCL5 expressed by MuSC in the dKO are the driver for the recruitment of CCR5 expressing MPs.

A: Thanks for the critical comment. As stated in the answer to your comment 3.11 (page 11 of this file), we don't think there is a CCR5+ MP subtype. As stated in the answer to your comment 3.8 (page 10 of this file), we agree that other cells such as T cells, MPs and muscle fibers also

express *Ccl5*. In fact, the main message from this figure is that MVC can be a potential pharmacological drug to alleviate dystrophy pathology. We have revised the writing on pages 16 and 21 to clarify the point.

3.14 Figure 5B: the timepoint of sampling for the MVC treatment was 2.5M after TMX injection. The CSA of the Ctrl mice receiving DMSO was around 2000 μm^2 . However, in Figure 1E, when the authors also quantified the CSA of the Ctrl muscle, the average CSA was around 1000 μm^2 . Why the average CSA of the Ctrl group (receiving vehicle control) so different in the two figures at the same timepoint?

A: Thanks for the critical comment. Reviewer 1 has a similar comment in his/her comment 1.9 (page 4 of this file), as stated in our answer to 1.9, we think the disparity may arise from the difference in the sex of the mice used in the experiments. Moreover, according to Reviewer 1's suggestions in his/her comment 1.15 (page 5 of this file), to enhance the accuracy of our assessment, we have now increased the number of experimental mice and conducted analyses of fiber size distribution, fiber count, and the percentage of fibers with centrally located nuclei (Fig. 5B-D, page 16). Altogether these results have solidified the MVC treatment improves muscle morphology and function.

3.15 Figure 5F-H: the authors showed that the relative abundances of the three populations were brought to comparable levels by CCR5 injection. What is the expression pattern of CCL5 in the dKO after MVC administration? Is TGF β 1 level reduced in the muscle?

A: Thanks for the excellent suggestions. We have now assessed the expression of CCL5 after MVC treatment and found the treatment did not affect the elevated expression of CCL5 in dKO MuSCs and TA muscles at both protein (Suppl. Fig. S5A, page 16) and RNA (Suppl. Fig. S5B-C, page 16) levels. We have also examined the TGF β 1 expression in the TA muscles after MVC treatment and found a significant reduction in MVC vs. DMSO-treated dKO muscles (Suppl. Fig. S5D-E, page 16).

3.16 Figure 6F: the authors proposed that YY1 does not directly bind to the promoter regions of CCL5, instead it can bind to an "enhancer" region. The genome track showed that the H3K27ac peaks seem to be at the promoter regions of the Taf15 gene? How does the authors reconcile this result? If the YY1 is mediating the transcription of Taf15 gene, which is the so-called enhancer region according to the authors, then the interpretations are totally different. Figure 6O-R then means that the Taf15 promoter was active in the dKO compared to the Ctrl.

A: Thanks for the critical comment. In fact, it is very common for one gene's enhancer to be another's promoter(5). YY1 may regulate both *Ccl5* and *Taf15* genes. However, we found no alteration in *Taf15* expression following YY1 deletion (Suppl. Table. S3), suggesting YY1 may not regulate *Taf15* gene expression. This was also confirmed by additional RT-qPCR assay in Ctrl and dKO MuSCs (below). Therefore, the binding of YY1 on this locus mainly functions to regulate *Ccl5*.

3.17 Figure 6T: as proposed in the manuscript, YY1 mediates the enhancer looping with promoter at the *Ccl5* locus. Enhancers also possess low level transcription capability, did the authors take this into account? In addition, how does EP looping work in a circular DNA vector? Would it be possible to use a YY1-expressing plasmid in the luciferase assay which allows the authors to better pinpoint if YY1 can drive the transcription?

A: Thanks for the interesting comments and suggestions. Indeed, it is well known that enhancers also possess low level transcription capability, generating so-called eRNAs(6, 7), but we reason this is not directly relevant to the current study which primarily elucidates how YY1 regulates *Ccl5* gene transcription. Regarding the second question, we agree that the E-P looping is not necessary in the context of a circular DNA vector due to the short distance. The reporter assay is commonly used to confirm the functionality of a regulatory element in activating or silencing a promoter. In our case, results from the assays proved that the identified *Ccl5* enhancer silences the promoter and it is under the regulation of YY1 protein. We have now revised the writing on page 19 to avoid misunderstanding. For the third suggestion, considering *Ccl5* expression in Ctrl cells is already very low, it will be very challenging to detect a further reduction upon YY1 over-expression. We therefore reason the loss of function assay in Fig. 6T is sufficient to prove YY1 drives the CCL5 transcription.

3.18 Figure 6U: what is the directionality of the Luciferase relative to the enhancer/promoter in the vector design? Depends on the design, this result can merely mean that the “enhancer” region can drive the transcription of the luciferase.

A: Thanks for the critical comment. We apologize for the incorrect illustration “E-L-P” in the original submission. We have now revised the illustration to “E-P-L”.

3. 19 The overall figure layout is often redundant or un-necessary, and the placement of the individual figures doesn’t follow a logical order. For example, Figure 3H showed that CCL5 is increased in the muscle, yet immediately after that comes the MP expression pattern, and then CCL5/CCR5 expression pattern in TA muscle again. Followed by an IF staining. The order of the images does not follow a good order. Another example is Figure 4M and 4N, why would the authors put the 2M data before the 7D data which showed no difference?

A: Thanks for the critical comment. We have now streamlined the figures and reorganized them

in a more coherent sequence. In particular, Fig. 3I-K has been reordered following your suggestion (please also see our answer to your comment 3.6, page 9 of this file). Fig. 4M-N have also been reordered with the original Fig. 2M incorporated into Fig. 4L and the 7D moved to Suppl. Fig. S4H.

3.20 Title misleading and should specific information such as “mouse model of Duchenne muscular dystrophy”

A: Thanks for the critical suggestion. We have now changed the title to “Skeletal Muscle Stem Cells Modulate Niche Function in Duchenne Muscular Dystrophy Mouse through a YY1-CCL5 Axis”.

3.21 There are many grammar mistakes throughout the manuscript.

A: Thanks for your comment. We have now carefully edited through the manuscripts to correct grammar mistakes.

3.22 Pp11, “while the stressed subset displayed a decrease (31% vs 23.2%)”, are the numbers in the wrong place since the comparison is dKO vs Ctrl?

A: Thanks for your comment. We apologize for the mistake which has now been corrected on page 11.

3.23 The role of MuSCs secreting immune modulatory cytokines has been documented in other studies (eg. Zhu et al., PMID: 27524611). Claiming this finding as “unappreciated” appear to be exaggerating.

A: Thanks for the critical comment. We have now reworded the writing on pages 2 and 20.

3.24 In Figure 2E, did the authors perform doublet-removal for the scRNA-seq analysis?

A: Thanks for the critical comment. A figure illustrating the process of doublet removal has now been included in Suppl. Fig. S2I and described on page 10.

3.25 Figure 3L-3N, what is the rationale of using mdx mice derived BMDM for the migration assay?

A: Thanks for the critical comment. BMDMs represent a scientifically viable option for investigating the impact on macrophage migration within a co-culture system(8, 9). *mdx* mice derived BMDMs were used to co-culture with Ctrl or dKO MuSCs in order to test their distinct effect on MPs.

3.26 Figure 4M, the staining of TGFb1 in the dKO group seems to be non-specific, especially the big bulge of the signal.

A: Thanks for the critical comment. We apologize for misusing the non-representative images in the original submission, they have been replaced by representative ones to show specific staining of TGF β 1 (Fig. 4L).

3.27 Figure 4F: what is the collection timepoint after EdU incorporation?

A: Thanks for the critical question. We apologize for the undetailed description in the original submission. The FAPs were collected 5 days after the TMX injection which has now been described in Fig. 4F.

3.28 Figure 7 legends too brief. There is no information connecting CCR5 to TGFbeta.

A: Thanks for the suggestion. We have now expanded the legend for Figure 7. But there may be a misunderstanding of the CCR5-TGF β 1 connection; this was not demonstrated in our study.

3.29 Discussion on FAP subset affected as different FAPs have distinct functions – some promotes regeneration.

A: Thanks for the critical suggestion. We have now included the discussion regarding the beneficial role of FAP subsets on pages 4-5.

Reference

1. F. Chen *et al.*, YY1 regulates skeletal muscle regeneration through controlling metabolic reprogramming of satellite cells. *EMBO J* **38**, (2019).
2. D. R. Lemos *et al.*, Nilotinib reduces muscle fibrosis in chronic muscle injury by promoting TNF-mediated apoptosis of fibro/adipogenic progenitors. *Nat Med* **21**, 786-794 (2015).
3. S. N. Oprescu, F. Yue, J. Qiu, L. F. Brito, S. Kuang, Temporal Dynamics and Heterogeneity of Cell Populations during Skeletal Muscle Regeneration. *iScience* **23**, 100993 (2020).
4. R. S. Blanc *et al.*, Inhibition of inflammatory CCR2 signaling promotes aged muscle regeneration and strength recovery after injury. *Nat Commun* **11**, 4167 (2020).
5. R. Andersson, A. Sandelin, Determinants of enhancer and promoter activities of regulatory elements. *Nature Reviews Genetics* **21**, 71-87 (2020).
6. Y. Zhao *et al.*, MyoD induced enhancer RNA interacts with hnRNPL to activate target gene transcription during myogenic differentiation. *Nature communications* **10**, 5787 (2019).
7. D. Wang *et al.*, Reprogramming transcription by distinct classes of enhancers functionally defined by eRNA. *Nature* **474**, 390-394 (2011).
8. E. S. Mullins, B. A. Konkle, C. E. McGuinn, W. Engl, S. D. Tangada, Efficacy and safety results from a phase 3b, open-label, multicenter, continuation study of rurioctocog alfa pegol for prophylaxis in previously treated patients with severe hemophilia A. *Blood* **132**, 2483 (2018).
9. L. Chen *et al.*, Neutrophil extracellular traps promote macrophage pyroptosis in sepsis. *Cell death & disease* **9**, 597 (2018).

REVIEWER COMMENTS

Reviewer #1 (Remarks to the Author):

After careful review of the rebuttal and the revised manuscript, we are not convinced that the manuscript meets the high standards of Nature Communications.

While the authors have made some progress in addressing our concerns, the data presented remain insufficiently novel and impactful to warrant publication. As we noted previously, the most compelling findings are those related to the CCR5 antagonist treatment. However, additional *in vivo* genetic evidence is still required to firmly establish the significance of these results.

Regarding the animal data, we remain concerned about the experimental design and the quality of the data presented. Specifically, the authors should have conducted a power analysis prior to their experiments to determine the appropriate sample size. The fact that they increased their sample size only after our review suggests a lack of careful planning. Furthermore, the changes in key outcomes with increased sample size raise questions about the reliability of the data.

Additionally, we are puzzled by the lack of variability in the control data for figures 1B, 1I, 1J, 3E, 3I, 3K, 4J, 4L, 4N, 6Q, and 6S. There is no explanation in the methods for how multiple controls can yield identical results with no standard deviation or standard error of the mean.

The authors' explanations for discrepancies in their control data, attributing them to the use of male and female mice, are worrisome. Given the well-established sex differences in muscle regeneration, the authors failed to provide clear information about the sex of animals used in each experiment in both the original manuscript and the revised version. This lack of transparency, combined with the low sample size, raises significant concerns about the validity of their data.

Reviewer #2 (Remarks to the Author):

The authors have adequately addressed the points raised in my previous review.

Reviewer #3 (Remarks to the Author):

The authors did a great job addressing my previous comments and provided further evidence to strengthen the Y1-CCL5 axis in the Dmd mouse model.

Reviewer #4 (Remarks to the Author):

Reviewer #5 (Remarks to the Author):

Point to point response to reviewer comments:

Reviewer #1

After careful review of the rebuttal and the revised manuscript, we are not convinced that the manuscript meets the high standards of Nature Communications.

1.1 While the authors have made some progress in addressing our concerns, the data presented remain insufficiently novel and impactful to warrant publication. As we noted previously, the most compelling findings are those related to the CCR5 antagonist treatment. However, additional in vivo genetic evidence is still required to firmly establish the significance of these results.

A: Thanks for the critical comment. We are sorry that the reviewer could not appreciate the novelty and impact of our study. As pointed out in both the original and revised manuscripts, we believe the novelty of our study is multiple. First, our study underscores the pivotal role of MuSCs in modulating the niche in dystrophic muscle, which was previously unappreciated. Second, we show that CCL5/CCR5 axis mediates the MuSCs/MPs crosstalk. Third, our study indicates that Maraviroc can be a potential therapeutic treatment approach for DMD. Lastly, we demonstrate YY1 represses Ccl5 transcription in MuSCs by directly binding to its enhancer thus facilitating promoter-enhancer looping. Nevertheless, we agree with the reviewer that in vivo genetic evidence could to some extent strengthen the function of CCL5-CCR5 axis. But we feel we have packed enough data into the current study and the suggested genetic study may not be absolutely necessary.

1.2 Regarding the animal data, we remain concerned about the experimental design and the quality of the data presented. Specifically, the authors should have conducted a power analysis prior to their experiments to determine the appropriate sample size. The fact that they increased their sample size only after our review suggests a lack of careful planning. Furthermore, the changes in key outcomes with increased sample size raise questions about the reliability of the data. Additionally, we are puzzled by the lack of variability in the control data for figures 1B, 1I, 1J, 3E, 3I, 3K, 4J, 4L, 4N, 6Q, and 6S. There is no explanation in the methods for how multiple controls can yield identical results with no standard deviation or standard error of the mean.

A: Thanks for the critical comment. We agree that power analysis before experiments is necessary for clinical studies. However, in some pioneering biological investigations, the absence of established reference data can pose challenges in predicting the anticipated outcomes essential for power calculations. We would like to point out that there were no changes in the key outcomes after the increased sample size during the first revision. There was only one minor change in Fig. 4F which did not alter the main conclusions of our study. We apologize for the puzzling explanation for the data in figures 1B, 1I, 1J, 3E, 3I, 3K, 4J, 4L, 4N, 6Q, and 6S. We have now revised the Methods (page 34) to clarify how the statistical analysis was conducted. Briefly, we performed paired Student's t test to assess statistical

significance in these figures, and it is commonly used in many publications including our recent ones (1-3).

1.3 The authors' explanations for discrepancies in their control data, attributing them to the use of male and female mice, are worrisome. Given the well-established sex differences in muscle regeneration, the authors failed to provide clear information about the sex of animals used in each experiment in both the original manuscript and the revised version. This lack of transparency, combined with the low sample size, raises significant concerns about the validity of their data.

A: Thanks for the critical comment. We apologize for un-intentionally omitting the information of the animal sex in the original submission which was clarified in the revised manuscript. We have now included the information on page 39. Despite extensive research on sex-specific disparities in muscle regeneration, our data in fact revealed no sex bias (both male and female mice showed similar dystrophic phenotypes following YY1 deletion). Moreover, for each animal experiment, mice of the same age and sex were used to ensure robust and consistent results. To increase transparency, we have now added a statement in Methods (page 24).

Reference

1. Y. Zhao *et al.*, Multiscale 3D genome reorganization during skeletal muscle stem cell lineage progression and aging. *Science Advances* **9**, eabo1360 (2023).
2. X. Chen *et al.*, Translational control by DHX36 binding to 5' UTR G-quadruplex is essential for muscle stem-cell regenerative functions. *Nature Communications* **12**, 5043 (2021).
3. Y. Qiao *et al.*, Nuclear m6A reader YTHDC1 promotes muscle stem cell activation/proliferation by regulating mRNA splicing and nuclear export. *Elife* **12**, (2023).

REVIEWERS' COMMENTS

Reviewer #3 (Remarks to the Author):

The revised manuscript addressed residual comments from Reviewer 1. In my opinion they did a good job responding to these comments